# Glia instruct axon regeneration via a ternary modulation of neuronal calcium channels in *Drosophila*

Shannon Trombley[1,4], Jackson Powell [1,4], Pavithran Guttipatti [1,4], Andrew Matamoros[1,2], Xiaohui Lin[3], Tristan O'Harrow [1], Tobias Steinschaden [1], Leann Miles [1], Qin Wang[1,2], Shuchao Wang[1], Jingyun Qiu[1], Qingyang Li[3], Feng Li[3] ✉ & Yuanquan Song [1,2]✉

A neuron's regenerative capacity is governed by its intrinsic and extrinsic environment. Both peripheral and central neurons exhibit cell-type-dependent axon regeneration, but the underlying mechanism is unclear. Glia provide a milieu essential for regeneration. However, the routes of glia-neuron signaling remain underexplored. Here, we show that regeneration specificity is determined by the axotomy-induced $Ca^{2+}$ transients only in the fly regenerative neurons, which is mediated by L-type calcium channels, constituting the core intrinsic machinery. Peripheral glia regulate axon regeneration via a three-layered and balanced modulation. Glia-derived tumor necrosis factor acts through its neuronal receptor to maintain calcium channel expression after injury. Glia sustain calcium channel opening by enhancing membrane hyperpolarization via the inwardly-rectifying potassium channel (Irk1). Glia also release adenosine which signals through neuronal adenosine receptor (AdoR) to activate HCN channels (Ih) and dampen $Ca^{2+}$ transients. Together, we identify a multifaceted glia-neuron coupling which can be hijacked to promote neural repair.

Advances in molecular biology have helped uncover the extrinsic environmental and intrinsic neuronal coordination required for axon regeneration[1–5]. Regeneration potential varies across types of neurons, including subtypes of retinal ganglion cells and dorsal root ganglion somatosensory cells[6,7]. Understanding the external and internal differences dictating regeneration specificity will help identify therapeutics capable of boosting the regeneration of otherwise stagnant neurons. Our prior work has documented the cell-type-specific regeneration in the dendritic arborization (da) sensory neurons of *Drosophila melanogaster*[8]. Da sensory neurons tile the body wall of larvae, are grouped into four classes based on morphology, and show

stereotyped patterning and localization[9]. The axon and soma of the da neurons are wrapped by layers of glial cells[10]. Class IV da sensory (C4da) neurons in fly larvae serve as nociceptors that detect harsh touch and heat[11], while Class III da (C3da) sensory neurons are mechanosensory for gentle touch[12]. We previously established the larval da neuron injury model where single axons of C4da or C3da neurons can be injured in vivo using a 2-photon laser and tracked over time for regeneration[8,13,14]. While C4da neurons regrow axons after peripheral injury, C3da neurons fail to do so[8].

$Ca^{2+}$ is known to play an important role in regeneration[15]. While $Ca^{2+}$ influx upon axonal or glial injury aids in relaying a retrograde

[1]Raymond G. Perelman Center for Cellular and Molecular Therapeutics, The Children's Hospital of Philadelphia, Philadelphia, PA 19104, USA. [2]Department of Pathology and Laboratory Medicine, University of Pennsylvania, Philadelphia, PA 19104, USA. [3]Department of Neurosurgery, Zhongshan Hospital, State Key Laboratory of Medical Neurobiology, MOE Frontiers Center for Brain Science, Institute for Translational Brain Research, Fudan University, 200032 Shanghai, China. [4]These authors contributed equally: Shannon Trombley, Jackson Powell, Pavithran Guttipatti. ✉e-mail: feng@fudan.edu.cn; songy2@chop.edu

injury signal, triggering vesicle exocytosis, transcriptional changes, and axon regeneration[15–17], an inhibitory role of neuronal Ca²⁺ currents has also been proposed[18,19], indicative of the unresolved complexity of Ca²⁺ signaling. We found that axotomy-induced global Ca²⁺ transients, mediated by L-type voltage-gated calcium channels (VGCCs), propels regeneration. VGCCs permit cytosolic Ca²⁺ influx and in mammals are typically formed of multiple subunits: an α1 transmembrane pore-forming subunit, an intracellular β subunit, an extracellular α2 subunit that is linked to a δ subunit, and a transmembrane γ subunit[20]. Multiple forms of each subunit exist in mammals. In *Drosophila*, there are three different α1 subunits (Ca- α1T, Cac, Ca- α1D), one β subunit (Ca-β), three linked α2δ subunits, and a possible γ subunit[21,22]. Here, we utilized the C4da and C3da neuron axon injury model to characterize the role of the Ca-α1D and Ca-β subunits in axon regeneration, as these were shown to influence both regeneration and Ca²⁺ transients. Through multiplexed fluorescent in situ hybridization (RNAscope), we found that the quantity and ratio of Ca-α1D to Ca-β expression varies with cell-type. Modulation of this ratio allows for control of regeneration ability and specificity.

After injury, glia and immune cells are also activated to produce an injury response. The expression of the inflammatory cytokine tumor necrosis factor-α (TNF-α) is transiently elevated in mammals after spinal cord injury[23]. The effect of TNF-α on regeneration and functional recovery is mixed in the current literature. TNF-α released from macrophages after spinal cord injury in zebrafish regulates the immune response and is necessary for regeneration[24], and application of TNF-α promotes mammalian optic nerve regeneration[25]. However, glia-derived TNF-α has also been shown to be inhibitory for neurite outgrowth in vitro[26]. Along these lines, treatment with a TNF-α antagonist has been shown to improve regeneration after peripheral injury[27], and TNF-α null mice show enhanced locomotor recovery after spinal cord injury[28]. We find here that the fly homolog of TNF-α, eiger (egr), is released from glia and acts through the fly TNF-α receptor, wengen (wgn), on neurons to control the expression of the VGCC subunits in neurons.

The growth-supportive role of glia in the peripheral nervous system (PNS) is well-established. Nerve injury triggers the conversion of myelinating and non-myelinating Schwann cells to an immature state to promote repair by downregulating myelin genes, upregulating trophic factors, enhancing the innate immune response, forming the Bungner's bands and guiding regrowing axons[29,30]. Satellite glial cells, which envelop the neuronal soma, have also been reported to promote regenerative growth[31]. Furthermore, the electrical properties of neurons are influenced by glial cells that wrap neuronal cell bodies and axons, and express their own ion channels. While many biochemical signaling mechanisms between neurons and glia have been identified, how the electrical effect of glia influences regenerating axons is less well studied. In the mouse brain, astrocytic glia were shown to sustain burst firing of neurons under depression, via hyperpolarization-dependent T-type VGCC de-inactivation, through increased glial expression of an inwardly-rectifying potassium channel (Kir4.1)[32,33]. Also in mice, glial purinergic signaling was shown to modulate neuron excitability through adenosine receptor's (A₂ₐR) interaction with hyperpolarization-activated cyclic nucleotide-gated (HCN) channels[34]. HCN channels have been shown to prolong steady-state inactivation of VGCCs through membrane depolarization[35]. Due to mixed literature on neural activity after injury, we asked whether a similar neuron-glia ion channel interaction occurs and could affect the neuronal regenerative potential. In *Drosophila*, the glia wrapping da neuron axons most closely resemble the Remak bundles formed by non-myelinating Schwann cells in mammals, with multiple axons being individually surrounded in a single bundle[36].

In this work, we identify a glial inwardly-rectifying potassium channel, *Irk1*, that works in concert with the calcium channel subunits in neurons to promote Ca²⁺ transients and thus regeneration after injury, and an inhibitory purinergic signaling through neuronal AdoR acting on the same calcium channels. Our work thus provides a multicellular picture of the ion channel machinery controlling regeneration differences between neuron subtypes and provides insights into how glial cells shape neuronal regeneration capacity.

## Results

### Neuronal calcium channel subunits dictate regeneration cell-type specificity

We first sought to understand what principles allow C4da, but not C3da neurons, to regenerate after injury, and hypothesized that it may be due to differences in electrical activity. To investigate this, we developed a method of calcium imaging which allowed us to monitor activity in unanesthetized fly larvae in vivo (Supplementary Fig. 1a, "Methods"). Ca²⁺ activity in C4da and C3da neurons was visualized with the membrane-targeted genetically-encoded calcium indicator myr::GCaMP6 and the LexA/LexAop system[37–39]. While uninjured C4da neurons in wild-type (WT) showed relatively constant low Ca²⁺ levels, axotomy induced strong transients throughout the soma and processes (Fig. 1a, b and Supplementary Movie 1), with 0% of uninjured and 65% of injured neurons spiking at 24 h after injury (h AI) (Fig. 1c). The total number of spikes was counted to determine the spiking rate, with injured C4da neurons showing an average of 0.51 ± 0.09 spikes/min (Fig. 1d). Importantly, the Ca²⁺ transients were specifically induced after axotomy, but not after solely injuring a single or all dendrites of C4da neurons (Supplementary Fig. 1b–i). We next examined the calcium activity of C3da neurons. Given that C3da neurons are mechanosensitive and could functionally spike in response to larval movements during imaging, only spikes that occurred without any larval movement were counted in the analysis (Methods). Uninjured C3da neurons did not show Ca²⁺ transients without movement, and in contrast to C4da neurons, only 10% of injured C3da neurons showed spiking (Fig. 1a–c and Supplementary Movie 2), with the average spiking rate being 0.04 ± 0.03 spikes/min (Fig. 1d). Based on this finding, we performed a genetic screen of calcium channels to identify the source of the Ca²⁺ transients in C4da neurons. C4da neuron-specific knockdown of *Ca-α1D* and *Ca-β*, L-type VGCC subunits, but not *Ca-α1T*, curtailed axon regeneration (Fig. 1e, f and Supplementary Fig. 1j, and see Methods for quantifications) and Ca²⁺ transients (Fig. 1g–j). The defects in regeneration and Ca²⁺ transients are further confirmed in *Ca-α1D* loss of function (LoF) mutants (Fig. 1e–j). To further test the necessity of Ca²⁺ transients in mediating axon regeneration, we sought to inhibit the Ca²⁺ transients by decreasing membrane depolarization via overexpressing Kir2.1[40], which significantly impeded axon regeneration (Supplementary Fig. 1n, o).

We then wondered if contrasting expression of these calcium channel subunits among C3da and C4da neurons is responsible for the cell-specific regeneration patterns. Our RNAscope analysis showed that in uninjured neurons, *Ca-α1D* is differentially expressed among subtypes, high in C4da neurons (labeled with *ppk-CD4tdGFP*) but low in C3da neurons (labeled with *19-12-Gal4>CD4tdGFP, repo-Gal80*), while *Ca-β* levels were comparable (Fig. 2a and Supplementary Fig. 2a). The fidelity of the RNAscope method in flies was verified by a positive control with a probe for *GFP*, a negative control using a probe against *Bacteria RNA*, positive controls comparing mRNA of WT and *Ca-α1D* or *Ca-β* overexpression in C3da neurons, and negative controls comparing mRNA of WT and *Ca-α1D* or *Ca-β* knockdown using a heterozygous deficiency plus an RNAi targeted to C4da neurons (Supplementary Fig. 2b–h). C3da neurons exhibited drastically reduced levels of both *Ca-α1D* and *Ca-β* after axon injury. In injured C4da neurons, *Ca-α1D* was also reduced but a significant amount remained, whereas *Ca-β* appeared largely unaltered (Fig. 2a, b and Supplementary Fig. 2a). We verified the expression of Ca-α1D with two antibodies. Using a published Ca-α1D antibody[41], we found that Ca-α1D protein was present in the soma, axon and dendrites of C4da neurons and it was

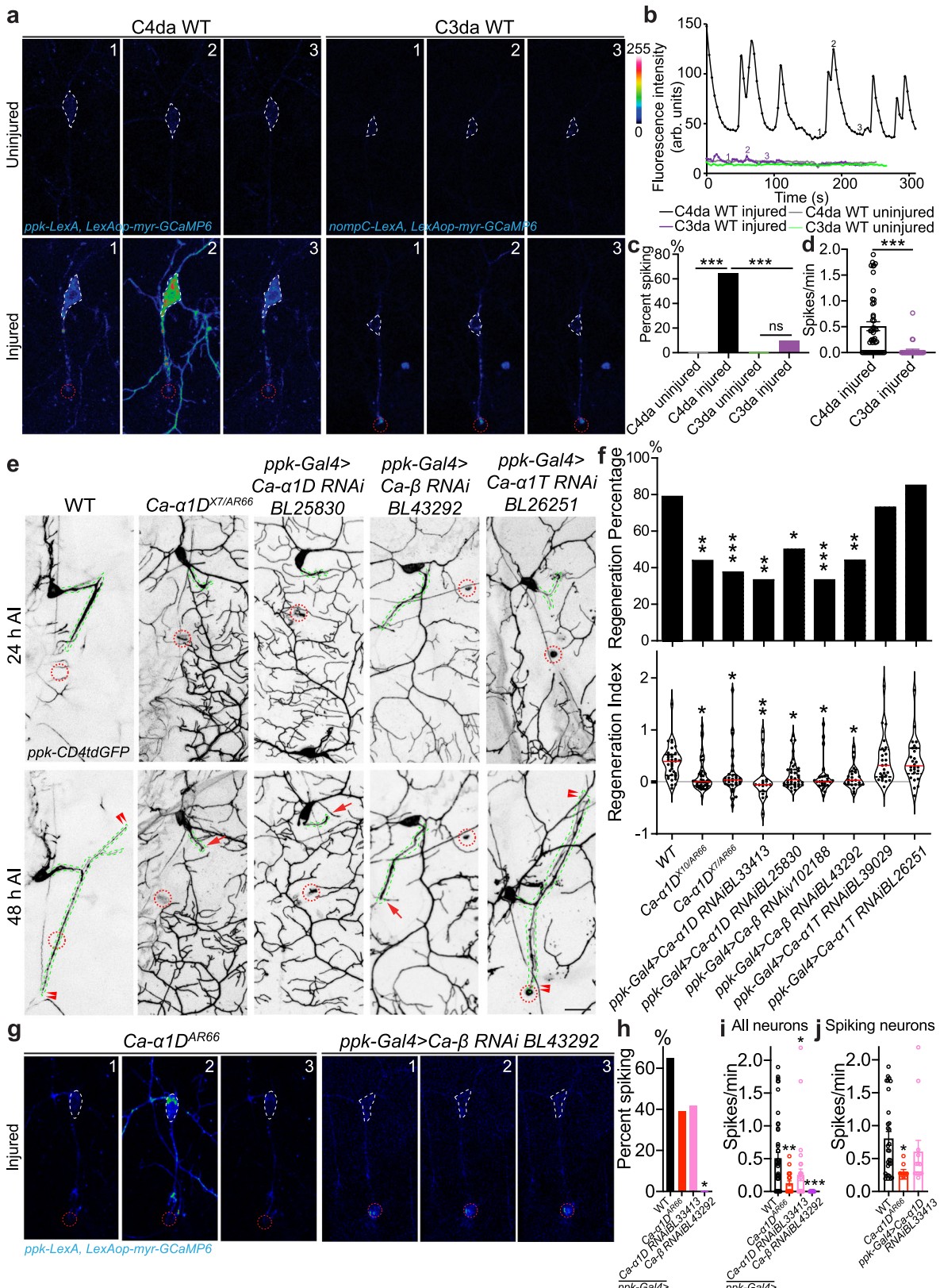

slightly but significantly more abundant in C4da neuron soma than C3da neurons (Supplementary Fig. 3a, b). Moreover, its expression was reduced after axotomy (Supplementary Fig. 3a, b), consistent with our RNAscope result. We also generated a polyclonal antibody for Ca-α1D, which showed obvious expression in C4da neurons (Supplementary Fig. 3k). The specificity of this antibody was confirmed by the reduced

staining in C4da neurons with *Ca-α1D* knockdown using two independent RNAis and elevated staining in C3da neurons with cell-type specific overexpression (Supplementary Fig. 3c, d, g, h). We also generated a polyclonal antibody against Ca-β, which showed obvious expression in C4da neurons. The specificity of this antibody was confirmed by the reduced staining in C4da neurons with *Ca-β* knockdown

**Fig. 1 | Axon regeneration depends on neuronal axotomy-induced Ca²⁺ transients. a–d** C4da, but not C3da neurons, generate Ca²⁺ transients after axotomy. **a** C4da neurons exhibit axotomy-induced Ca²⁺ transients throughout the cell, whereas C3da do not. The cell body is outlined with white dashed line and the injury site is marked by red dashed circle. **b** Plots of the mean fluorescence intensity over time in the cell body. Uninjured C4da neurons, uninjured and injured C3da neurons show minimal changes in Ca²⁺ levels. **c** Quantification of the percent of neurons showing Ca²⁺ spikes. $N = 12, 48, 6, 30$ neurons. $P > 0.9999, <0.0001, >0.9999, <0.0001$. **d** Quantification of the number of Ca²⁺ spikes per minute. $N = 48, 32$ neurons, average values and standard error of the means are shown. $P = <0.0001$. **e** $Ca$-$\alpha 1D$ and $Ca$-$\beta$ are required for C4da axon regeneration. The injury site is demarcated by the red dashed circle, and axons are traced by a green dashed line. Arrow marks axon stalling while arrowheads show the regrowing axon tips.

**f** Quantifications of C4da neuron axon regeneration with regeneration percentage and regeneration index. $N = 34, 41, 32, 18, 38, 27, 25, 26, 26$ neurons. $P = 0.0022, 0.0010, 0.0020, 0.0138, 0.0005, 0.0067, 0.7590, 0.7419$ (top). **g** Axotomy-induced Ca²⁺ transients in C4da neurons are reduced in $Ca$-$\alpha 1D$ mutants, and after C4da neurons knockdown of $Ca$-$\alpha 1D$ or $Ca$-$\beta$. **h–j** Quantification of the percentage of C4da neurons showing Ca²⁺ spikes (**h**), the spiking rate per minute in all neurons, (**i**) or neurons showing spiking. (**j**). $N = 48, 23, 31, 25$ neurons, $P = 0.0724, 0.0643, >0.0001, P = 0.0046, 0.0443, <0.0001$ (**h, i**) and $30, 9, 13$ neurons, $P = 0.0321, 0.4587$ (**j**), average values and standard error of the means are shown. $*P < 0.05$, $**P < 0.01$, $***P < 0.001$, Fisher's exact test, two-sided (**c, f** top and **h**), unpaired Student's $t$ test, two-tailed (**d**), one-way ANOVA followed by Dunnett's test (**f** bottom), Holm–Sidak's test (**c, i, j**). Scale bar = 20 μm. Source data are provided as a Source data file.

using two independent RNAis and elevated staining in C3da neurons with cell-type specific overexpression (Supplementary Fig. 3e, f, i, j). Collectively, these results led to our conclusion that $Ca$-$\beta$ is specifically maintained in C4da neurons upon injury, and that the ratio of $Ca$-$\beta$/$Ca$-$\alpha 1D$ is increased from 0.5 to 1 due to the selective reduction of $Ca$-$\alpha 1D$ (Fig. 2c), which may be essential for the emergence of Ca²⁺ transients after injury. This hypothesis is supported by a previous report that Ca-β modulates Ca-α1D channel activity by slowing down its inactivation[42].

To further test the hypothesis that an optimal Ca-β/Ca-α1D ratio may be a prerequisite for axon regeneration, we manipulated Ca-α1D and Ca-β expression in both C3da and C4da neurons. We found that overexpression of Ca-β or Ca-α1D alone in C3da neurons, which normally have low levels of Ca-β and Ca-α1D after injury, modestly enhanced their axon regeneration (Fig. 2d, e and Supplementary Fig. 1k, and see "Methods" for quantifications). Strikingly, overexpression of Ca-α1D + Ca-β led to stronger regeneration potential in C3da neurons than in the neighboring C4da neuron (ddaC)[8] (Fig. 2d, e and Supplementary Fig. 1k, m), while overexpression of a generic control ion channel TrpA1 at its non-activating temperature did not increase regeneration (Supplementary Fig. 2i, j). Contrasting to C3da, overexpression of Ca-β and/or Ca-α1D in C4da neurons did not improve their regeneration (Fig. 2f). Ca-α1D overexpression, on the contrary, reduced regeneration, confirming that lowering Ca-β/Ca-α1D ratio impairs regeneration (Fig. 2f and Supplementary Fig. 3l, m).

We then went on to determine if C3da neurons overexpressing Ca-α1D + Ca-β are capable of producing axotomy-induced Ca²⁺ transients similar to C4da neurons. We focused on C3da neurons overexpressing Ca-α1D + Ca-β. To our surprise, we did not observe an increased spiking rate after overexpression in still neurons (Fig. 2g, i). We also analyzed Ca²⁺ spikes associated with C3da neurons exhibiting a jittering behavior (slight back-and-forth movement with displacement <20 μm) —jittering-associated spikes (JAS). We found that the percent of jittering C3da neurons showing Ca²⁺ spikes was trending increased (Fig. 2j). Most strikingly, we found that the Ca²⁺ baseline in the soma after axotomy doubled after overexpression (Fig. 2h), and that more than half of the injured C3da neurons showed subthreshold transients (STT) rarely seen in WT (Fig. 2k, l and Supplementary Movie 3, "Methods"). Overexpression of Ca-α1D or Ca-β alone also slightly increased STT, but to a lesser extent than the co-overexpression (Supplementary Fig. 2k–m). These results suggest that Ca²⁺ spikes per se may not be required for axon regeneration, and that elevated baseline and/or subthreshold Ca²⁺ transients by Ca-α1D + Ca-β overexpression is sufficient for activating downstream signaling to drive regeneration.

### Axotomy-induced Ca²⁺ transients depend on peripheral glia

Glial cells are indispensable for axon regeneration in the PNS and we showed previously that C4da neurons failed to regenerate if the accompanying glia were simultaneously damaged[8]. We therefore asked if glial cells are also required for the Ca²⁺ transients after injury. We performed calcium imaging of C4da neurons together with glial cell labeling (with *repo-Gal4>mRFP*). We found that in the control

injury paradigm, where glial processes remained largely intact and still wrapped the C4da neuron soma and axon, Ca²⁺ spikes were robustly induced after axotomy (Fig. 3a–e and Supplementary Fig. 4a, b). However, when we injured the glial processes in addition to axotomy by targeting the glial processes or the glial nucleus (Supplementary Fig. 4a), resulting in unwrapping of the soma and the injured axon, Ca²⁺ transients were largely eliminated (Fig. 3a, b and Supplementary Fig. 4b, c). Fewer than 30% of these C4da neurons showed spiking, with a reduced spiking rate, compared to the tissue injury control as well as the control injured (Fig. 3a–d and Supplementary Fig. 4a, b). The spiking rate from only the spiking neurons was also reduced (Fig. 3e). This result confirms that glial wrapping integrity is indeed necessary for injury-induced Ca²⁺ transients, suggesting that glia support axon regeneration at least in part by mediating the electrical coupling. Speaking to this interplay after axotomy, we also noticed that glial processes around the regenerating axon appeared to undergo remodeling. Wrapping glia (labeled by *nrv2-Gal4*), a subset of glia which exclusively wrap the da neuron axons[10], changed from their even distribution surrounding uninjured axons to a clustered appearance after axotomy and occasionally extended thin processes that contacted the axon tip (Supplementary Fig. 5a, arrows and arrowheads). We speculate that this type of remodeling may lead to the formation of a specialized glia-neuron interaction/junction/sealing important for neuronal Ca²⁺ transients, which warrants further investigation. It is also possible that these glial processes represent a potential glial repair response that may be important for their modulation of axon regeneration.

### Glia-neuron TNF signaling is required for neuronal Ca²⁺ transients and regeneration

In search of glia-neuron interactions potentially capable of supporting peripheral axon regeneration, we came upon TNF-α signaling, which has been reported as a prodegenerative mechanism at the neuromuscular junction in flies[43]. TNF signaling is implicated in regeneration, but mostly in the context of the immune response[23–28]. We found that deficiency of *Drosophila* TNF-α, *egr*, in a LoF mutant[44] *egr³* severely impaired C4da neuron axon regeneration, presenting a regeneration index of $-0.10 \pm 0.04$ which is indicative of substantial axon stalling or retraction (Fig. 3f, g). Glia-specific knockdown of *egr* (*repo-Gal4>egr RNAiv45253*) recapitulated the phenotype (Fig. 3f, g), indicating its glial origin. Similarly, C4da neuron-specific knockdown of the fly TNF-α receptor (TNFR), *wgn*, with two independent RNAis (*ppk-Gal4>wgn RNAiv330339, ppk-Gal4>wgn RNAiv9152*) and *wgn* LoF in the *wgn²²* mutants[45] also decreased regeneration (Fig. 3f, g). LoF of the adapter protein TNF-receptor-associated factor 6 (Traf6), which is thought to mediate TNF signaling[46], also reduced regeneration (Fig. 3f, g). Notably, C3da neuron-specific overexpression of wgn was sufficient to convert them into regeneration-competent (Fig. 3h, i). Using the *egr-Gal4* reporter[43], we found that egr expression was restricted to glial cells, particularly along the da neuron axons, and that its transcription was upregulated after axotomy (Fig. 3j, k and

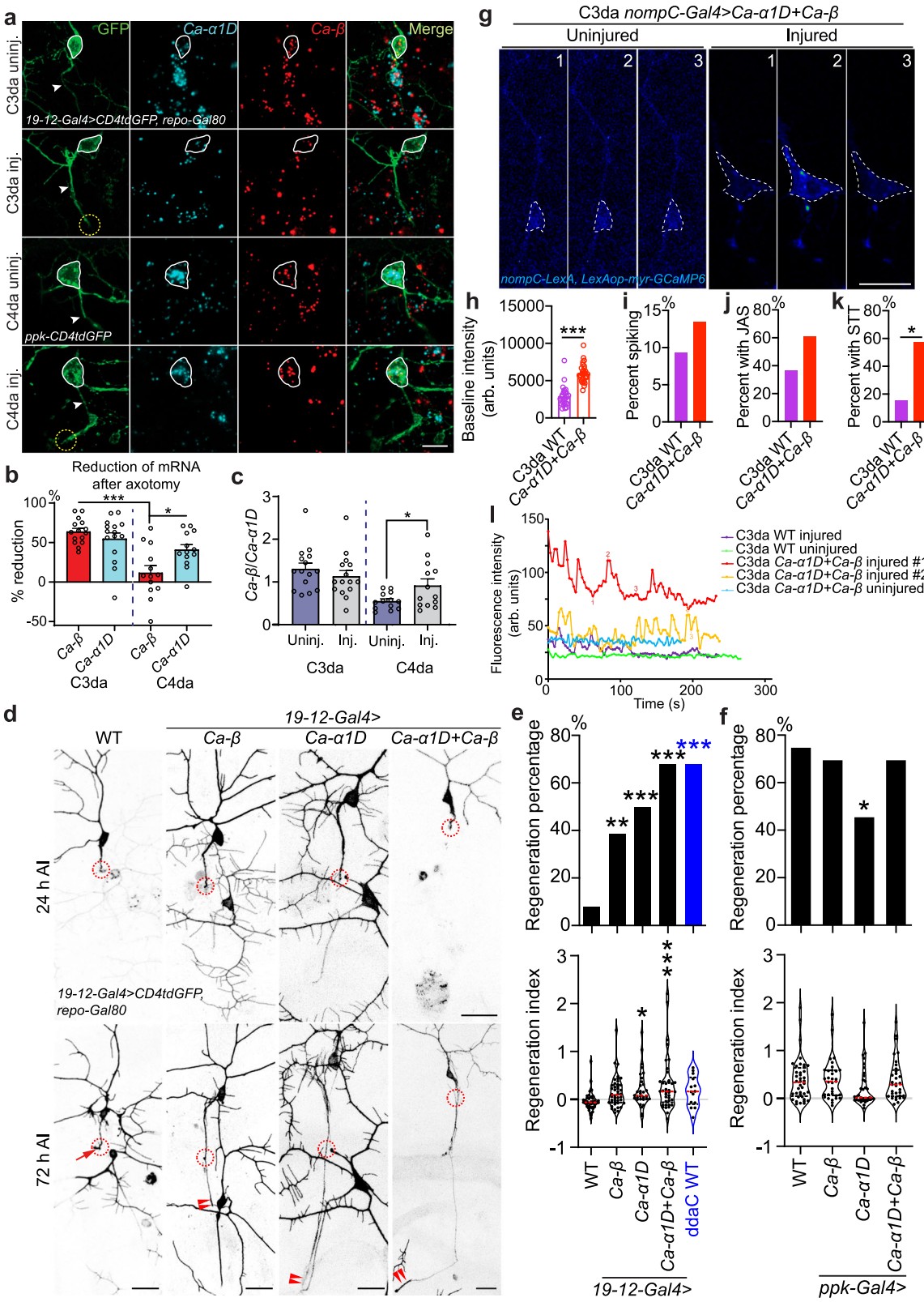

Supplementary Fig. 4d, e). Furthermore, we examined the expression pattern of wgn using a transgenic fly that contains a fosmid clone of the GFP-tagged wgn genomic locus, $wgn^{fTRG00673.sfGFP-TVPTBF}$, so that GFP expression reflects the endogenous pattern of wgn at the physiological level[47]. We found that wgn was present in both C4da and C3da neurons, with a slight but significant higher expression in C4da neurons (Fig. 3l, m), and that its expression remained unchanged

after axotomy (Fig. 3n). These data demonstrate that the glia–neuron egr–wgn ligand–receptor axis is both necessary and sufficient for promoting axon regeneration in the PNS. As a control, we examined if disrupting egr-wgn signaling affects da neuron morphology or glial wrapping. We found that glial egr RNAi or $wgn^{22}$ mutants did not impair the overall C4da neuron soma glial wrapping (Supplementary Fig. 5b, c), or grossly change the number of primary or secondary

**Fig. 2 | The ratio of Ca-β/Ca-α1D is essential for axon regeneration.**
**a**–**c** Expression analysis of *Ca-α1D* and *Ca-β* by RNAscope. **a** Without injury, *Ca-α1D* is highly expressed in C4da but low in C3da neurons, while *Ca-β* is similarly expressed in both cell types. After axotomy, in C3da neurons, both *Ca-α1D* and *Ca-β* are reduced. In C4da neurons, *Ca-α1D* is reduced but a significant level remains, whereas *Ca-β* remains unchanged. The cell body is outlined with the white circle and the dashed circle marks the injury site. Arrowhead marks the axon. **b** The percent of reduction for *Ca-α1D* and *Ca-β* after injury. *Ca-β* in C4da neurons show minimal reduction. $N = 15$ neurons, $P = < 0.0001, 0.0198$, average values and standard error of the means are shown. **c** The ratio of *Ca-β/Ca-α1D* with or without injury. $N = 15, 15, 13, 13$ neurons, $P = 0.0100$, average values and standard error of the means are shown. **d** C3da neuron overexpression of Ca-β and/or Ca-α1D increases axon regeneration. The injury site is demarcated by the dashed circle. Arrow marks axon stalling while arrowheads show the regrowing axon tips. **e**, **f** Quantifications of C3da (**e**) and C4da neuron (**f**) axon regeneration with regeneration percentage and regeneration index. $N = 38, 51, 30, 38, 25$ neurons, $P = 0.0048, 0.0002, <0.0001, <0.0001$ (top) (**e**) and 40, 33, 35, 33 neurons,

$P = 0.7926, 0.0168, 0.7926$ (top) (**f**). **g**–**l** C3da neurons overexpressing Ca-α1D and Ca-β show elevated $Ca^{2+}$ baseline and frequent subthreshold transients (STT) after axonal injury. Images from three timepoints (1, 2 and 3 shown in (**l**)) are presented. The dashed white circle marks the cell body. **h** C3da neurons with Ca-α1D + Ca-β overexpression show elevated baseline intensity than WT. $N = 25, 28$ neurons, $P = < 0.0001$. **i** C3da neurons with Ca-α1D + Ca-β overexpression do not show increased $Ca^{2+}$ spiking in neurons without movement. $N = 32, 37$ neurons, $P = 0.7160$. **j** C3da neurons with Ca-α1D + Ca-β overexpression show a trend of increased percent of jittering neurons with $Ca^{2+}$ spikes (JAS). $N = 30, 18$ neurons, $P = 0.1380$. **k** C3da neurons with Ca-α1D + Ca-β overexpression show increased percent of neurons with subthreshold $Ca^{2+}$ transients (STT). $N = 13, 26$ neurons, $P = 0.0173$. **l** Plots of the mean fluorescence intensity over time in the cell body. *$P < 0.05$, **$P < 0.01$, ***$P < 0.001$, Fisher's exact test, two-sided (**e** top, **f** top, **i**, **j**), unpaired Student's *t* test, two-tailed (**c**, **h**, **k**), one-way ANOVA followed by Tukey's test (**b**), Holm–Sidak's test (**e** bottom and **f** bottom). Scale bar = 20 μm. Source data are provided as a Source data file.

---

dendritic branches of C4da neurons (Supplementary Fig. 5b–e), but they shifted the complexity of dendrite patterning (Supplementary Fig. 5b, f). These data suggest that egr-wgn signaling likely regulates axon regeneration independent of its modulation of neuronal morphology.

To gain mechanistic insights into egr-wgn's function in regeneration, we tested its potential role in mediating the axotomy-induced $Ca^{2+}$ transients. We found that glia-specific knockdown of *egr* (*repo-Gal4>egr RNAiv45253*) and C4da neuron-specific knockdown of *wgn* (*ppk-Gal4>wgn RNAiv330339*) led to impaired $Ca^{2+}$ transients after injury. Both reduced the percent of C4da neurons showing $Ca^{2+}$ spikes and the overall spiking rate, as well as the spiking rate in only the spiking neurons (Fig. 4a–e). These results suggest that egr released from glial cells acts on neuronal wgn receptors to maintain $Ca^{2+}$ transients, and thus promote regeneration.

### TNF signaling maintains Ca-α1D expression after injury
To determine how egr-wgn signaling regulates $Ca^{2+}$ transients, we focused on the expression of the VGCC subunits. RNAscope analysis showed that removal of egr (in *egr³* mutants) or wgn (via C4da neuron-specific RNAi) led to a ~50% decrease of *Ca-α1D* mRNA transcripts in C4da neurons after axotomy, without affecting the expression of *Ca-β* (Fig. 4f–h). The uninjured levels of mRNA were not affected in the *egr³* background, but were elevated in the wgn RNAi (Supplementary Fig. 4f, g). Therefore, we overexpressed Ca-α1D specifically in C4da neurons in *egr³* mutants and found that it was capable of rescuing the reduction of regeneration caused by *egr* deficiency (Fig. 4i, j), confirming that Ca-α1D lies downstream of the egr–wgn axis, and that TNF signaling promotes axon regeneration by regulation of VGCC expression and axotomy-induced $Ca^{2+}$ transients.

### Glial inwardly rectifying potassium channels allow neuronal $Ca^{2+}$ transients and regeneration
C4da neurons show $Ca^{2+}$ spiking or bursts after injury and prior work in mice has demonstrated that bursting can arise from the interplay of VGCCs in neurons and inwardly-rectifying potassium channels, such as Kir4.1, in glia. Kir4.1 mediates glial removal of $K^+$ at the tight neuron–glia junction, decreases the $[K]_{out}$ and hyperpolarizes neuronal resting membrane potential, resulting in the de-inactivation of neuronal VGCCs to allow for repeated firing[32,33]. Given the role of both VGCCs and glia in $Ca^{2+}$ transients and regeneration, we asked if there is a glial inwardly rectifying potassium channel involved here as well. There are three Kir4.1 homologs in *Drosophila*: Irk1, Irk2, and Irk3. We thus investigated if glial Irks play a role in regeneration. Via glia-specific knockdown of Irks (with *repo-Gal4*), we found that Irk1 LoF (*repo-Gal4>Irk1 RNAiBL25823*) exhibited obvious reduction of axon regeneration in C4da neurons compared to WT (Fig. 5a, b), and was thus the

focus of our follow-up studies. This observation was confirmed by glial overexpression of a dominant-negative Irk1 with a mutated selectivity filter (Irk1-AAA)[48] (Fig. 5a, b). On the other hand, gain-of-function (GoF) of Irk1 via glial overexpression of Irk1-WT modestly but significantly increased the extent of axon regrowth in regenerating C4da neurons ddaC, without affecting the overall regeneration percentage (Supplementary Fig. 6a–c). These results suggest that glial Irk1 is both necessary and sufficient for promoting axon regeneration. Worth mentioning, glial knockdown of Irk1 did not grossly alter the glial wrapping and morphology of C4da neurons (Supplementary Fig. 5b–f).

Calcium imaging further showed that, compared to WT that exhibited multiple $Ca^{2+}$ spikes after axotomy, glia-specific knockdown of Irk1 and overexpression of the dominant-negative Irk1-AAA both significantly reduced the percentage of C4da neurons displaying spikes and the overall spiking rate (Fig. 5c–f and Supplementary Fig. 6d). The spiking rate from only the spiking neurons was also reduced (Fig. 5g). No obvious difference was detected for the uninjured neurons (Fig. 5c, d). Collectively, these results suggest that the $K^+$ buffering capacity of glial Irk1 channels is essential for maintaining the injury-induced $Ca^{2+}$ transients in C4da neurons and the subsequent axon regeneration.

RNAscope analysis showed that *Irk1* mRNAs were largely absent around the C4da neuron soma or proximal axon, but enriched as packets along the intermediate axon stretch overlapping with the domain covered by the wrapping glia (Fig. 5h, i, arrowheads). The distribution of *Irk1* mRNA remained similar after axotomy, and enrichment was observed in glial processes surrounding or ahead of the axon tip (Fig. 5h, i, asterisk). To confirm the expression of Irk1, we generated a polyclonal antibody against Irk1, which showed obvious expression in glial processes (Supplementary Fig. 6e). The specificity of this antibody was confirmed by the reduced staining in glial cells with glial *Irk1* knockdown using two independent RNAis (Supplementary Fig. 6e, f).

### Glia–neuron adenosine signaling suppresses axon regeneration through AdoR and the Ih ion channel
After identifying direct regulation of axon regeneration via membrane excitability, we became interested in the possibility of neuron membrane potential manipulation affecting VGCC activation. Glial release of ATP has been shown to affect neuron excitability through cAMP production by adenosine activated $A_{2a}R$, homologous to *Drosophila* adenosine receptor (AdoR). $A_{2a}R$ activates hyperpolarization-activated nucleotide gated (HCN) channels, homologous to inward hyperpolarization channel (Ih)[34]. We wondered if a similar mechanism modulates axon regeneration. We found that LoF mutant[49] *AdoR¹* and C4da neuron specific knockdown of *AdoR* with two RNAis (*ppk-Gal4>AdoR*

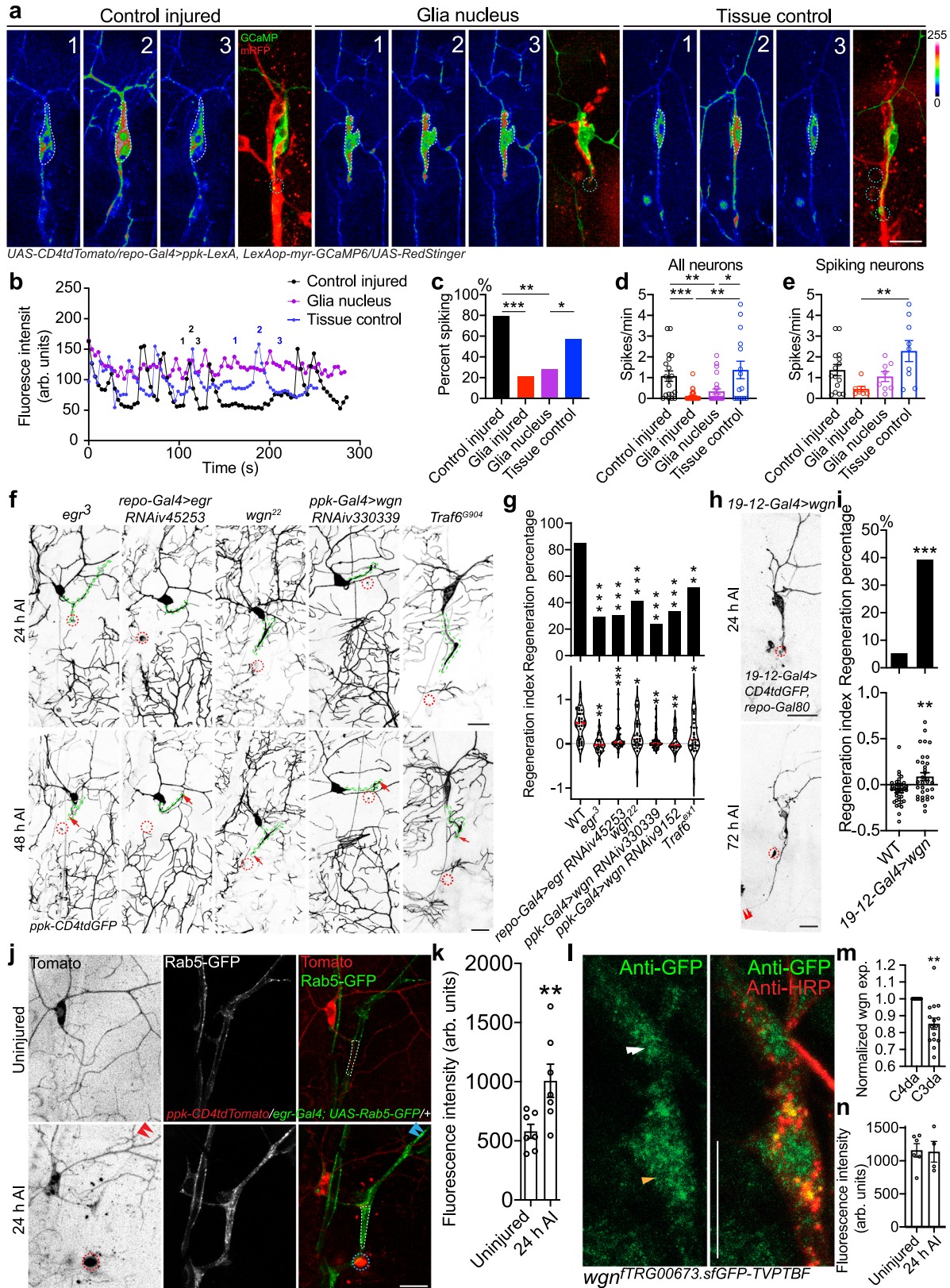

**a** Control injured | Glia nucleus | Tissue control

*UAS-CD4tdTomato/repo-Gal4>ppk-LexA, LexAop-myr-GCaMP6/UAS-RedStinger*

**b** Control injured · Glia nucleus · Tissue control

**c** Percent spiking

**d** All neurons    **e** Spiking neurons

**f** *egr³* | *repo-Gal4>egr RNAiv45253* | *wgn²²* | *ppk-Gal4>wgn RNAiv330339* | *Traf6^{G904}*

*ppk-CD4tdGFP*

**g** Regeneration index / Regeneration percentage

WT, egr³, repo-Gal4>egr RNAiv45253, wgn²², ppk-Gal4>wgn RNAiv330339, ppk-Gal4>wgn RNAiv9152, Traf6^{ex1}

**h** *19-12-Gal4>wgn*

*19-12-Gal4>CD4tdGFP, repo-Gal80*

**i** Regeneration percentage / Regeneration index

WT, 19-12-Gal4>wgn

**j** Tomato | Rab5-GFP | Tomato Rab5-GFP

*ppk-CD4tdTomato/egr-Gal4; UAS-Rab5-GFP/+*

**k** Fluorescence intensity (arb. units)

Uninjured, 24 h AI

**l** Anti-GFP | Anti-GFP Anti-HRP

*wgn^{fTRG00673.sfGFP-TVPTBF}*

**m** Normalized wgn exp.

C4da, C3da

**n** Fluorescence intensity (arb. units)

Uninjured, 24 h AI

*RNAiv1385* and *ppk-Gal4>AdoR RNAiBL56866*) significantly increased axon regeneration (Fig. 6a, b). *Ih* removal similarly boosted C4da regeneration with LoF mutant[50] *Ih^{f03355}* and C4da neuron knockdown (*ppk-Gal4>Ih RNAiBL58089* and *ppk-Gal4>Ih RNAiBL51765*) (Fig. 6a, b and Supplementary Fig. 7a, b). The regeneration enhancement in *Ih^{f03355}* mutant was suppressed by C4da neuron overexpression of an Ih

isoform (Ih.F; Fig. 6a, b). Transheterozygous mutants of *AdoR* and *Ih* (*Ih^{f03355/+}*; *AdoR^{1/+}*), but not individual heterozygous lines, showed an increase in regeneration, reconfirming they are in the same genetic pathway (Fig. 6c, d). Interestingly, glial knockdown of *AdoR* with two RNAis (*repo-Gal4>AdoR RNAiv1385* and *repo-Gal4>AdoR RNAi BL56866*), but not *Ih* (*repo-Gal4Ih RNAiBL58089* & *repo-Gal4>Ih*

**Fig. 3 | Glial TNF-α signaling is required for axotomy-induced Ca²⁺ transients and axon regeneration. a–c** Peripheral glial cells are required for axotomy-induced Ca²⁺ transients in C4da neurons. **a** C4da neurons exhibit Ca²⁺ spikes after axonal injury when the glial cells are not severely damaged. Images from three timepoints are presented. The white and teal dashed circles mark the cell body and injury site, respectively. C4da neurons and glia are labeled by GCaMP and mRFP, respectively. **b** Plots of the mean fluorescence intensity over time in the cell body. **c** Quantification of the percent of neurons showing Ca²⁺ spikes. **d, e** Quantification of the number of Ca²⁺ spikes per minute in all neurons or spiking neurons. N = 19, 33, 25, 14 neurons, P = < 0.0001, 0.0019, 0.0953, 0.0370, and P = < 0.0001, 0.0084, 0.0017, 0.0497 (**c, d**) and 15, 7, 8, 9 neurons, P = 0.0078 (**e**), average values and standard error of the means are shown. **f, g** LoF of egr in egr³ mutants or by glia-specific RNA, LoF of wgn by C4da neuron-specific RNAi and LoF of Traf6 in Traf6^{G904} mutants impair axon regeneration. Quantifications of C4da neuron axon regeneration with regeneration percentage (g top) and regeneration index (g bottom). N = 40, 24, 33, 29, 38, 24, 31 neurons, P = < 0.0001, <0.0001, <0.0001, <0.0001, 0.0002, 0.0036 (top). **h, i** C3da neuron overexpression of wgn increases axon regeneration. Quantifications of C3da neuron axon regeneration with regeneration percentage (**h**) and regeneration index (**i**). N = 37, 33 neurons, P = 0.0009 (top),

0.0033 (bottom), average values and standard error of the means are shown. **j** egr is expressed in glial cells wrapping the da neurons and its transcription is upregulated after axotomy. egr expression is assessed by egr-Gal4 driving a Rab5-GFP reporter. The injury site is demarcated by the dashed circle. Arrowheads show the regrowing axon tip. **k** Quantification of GFP fluorescence intensity in the area outlined by the white dashed line in (**j**). N = 7 neurons, P = 0.0070, average values and standard error of the means are shown. **l** wgn expression in C4da (orange arrowhead) and C3da (white arrowhead) neurons, assessed using a transgenic fly containing a fosmid clone of the GFP-tagged wgn genomic locus, wgn^{fTRG00673.sfGFP-TVPTBF}. Neurons are labeled with anti-HRP antibody. **m** Quantification of wgn expression in C4da and C3da neuron. Mean GFP fluorescence intensity in the soma is normalized to that of C4da neurons. N = 15 neurons, P = 0.0020, average values and standard error of the means are shown. **n** Fluorescence intensity of GFP in C4da neuron soma with or without axotomy. N = 6, 4 neurons, P = 0.9143, average values and standard error of the means are shown. *P < 0.05, **P < 0.01, ***P < 0.001, Fisher's exact test, two-sided (**c, g** top and **i** top), unpaired Student's t test, two-tailed (**i** bottom), one-way ANOVA followed by Holm–Sidak's test (**c, d, e, g** bottom). Mann–Whitney test, two-tailed (**k, n**), Wilcoxon test, two-tailed (**m**). Scale bar = 20 μm. Source data are provided as a Source data file.

---

RNAiBL51765) also showed increased regeneration (Supplementary Fig. 7c–f), suggesting a possible feedback with glial adenosine autocrine signaling independent of neuronal Ih. Neuronal knockdown of both showed mild but significant pro-regenerative phenotypes in C3da (19-12-Gal4>AdoR RNAiv1385 and 19-12-Gal4 > Ih RNAiBL51765) (Fig. 6e, f). Moreover, expression analysis using an AdoR protein trap AdoR-GFP (AdoR^{MI01202-GFSTF.1})[51,52] and AdoR-Gal4 knockin (AdoR^{2A-GAL4.KI})[53] showed that AdoR is expressed in both C4da and C3da neurons with or without axotomy (Fig. 6g). These results demonstrate that the AdoR-Ih axis inhibits axon regeneration in C4da and C3da neurons.

As membrane hyperpolarization de-inactivates Ca-α1D, we hypothesized Ih activation suppresses Ca²⁺ transients. Indeed, we found that C4da neuron specific knockdown of Ih (ppk-Gal4>Ih RNAiBL58089) significantly increased the rate of axotomy-induced Ca²⁺ transients in neurons that showed spiking, without changing the overall spiking percentage or rate (Fig. 6h–k). Ih knockdown not increasing the percentage of spiking neurons suggests the channels play a role only in attenuating the Ca²⁺ spikes of neurons with sufficient Ca-α1D and Ca-β levels and does not affect neurons incapable of spiking. The ability of HCN channels to limit Ca²⁺ spikes has been explored in other models and was shown to prolong the steady-state inactivation of VGCCs[35]. Our result suggests that the AdoR-Ih axis may inhibit axon regeneration via dampening Ca²⁺ transients.

As we have showed previously that v'ada, a C4da neuron, prefers to regenerate its axon dorsally to the cell body, away from its original direction[54], we asked if more preferable glia–neuron interactions occur on this side. Axon regeneration direction is graphically represented in (Supplementary Fig. 1l). Using iATPSnFR[55,56] to sense ATP in glia, we noticed that, while ATP level appears largely even across the glial processes along the axon tract in uninjured larvae, a gradient of ATP with higher concentration ventral to the cell body was readily visible at 24 and 48 h AI (Fig. 7a, b). However, mRFP driven to glia by repo-Gal4 did not show any gradient on the membrane or in the cytosol (Supplementary Fig. 8a, b), and the egr-Gal4 reporters may even exhibit a reverse gradient (Supplementary Fig. 8c, d), suggesting the specificity of the ATP gradient. This data led us to hypothesize that the ATP and hence adenosine gradient, low dorsal and high ventral, may induce a differential adenosine-AdoR signaling contributing to unfavored regeneration towards the ventral trajectory. To test this, we quantified the directionality of axon regeneration (see "Methods"[54]) and found that the percentage of ventrally regenerating axons is significantly increased in AdoR¹ mutants compared to WT, with a drastic increase of regeneration ratio indicative of robust ventral regrowth (Fig. 7d, e). AdoR knockdown in C4da neurons partially recapitulated the phenotype showing a similar trend (Fig. 7d, e). These data suggest that the

inhibitory glia–neuron adenosine signaling likely instructs axon regeneration by restricting overall regrowth and providing a guidance signal.

In order to confirm the glial origin of ATP and thus adenosine, we performed a genetic screen targeting the machinery responsible for ATP release and processing. We focused on Major Facilitator Superfamily Transporter 18 (MFS18), ortholog of SLC17A9 which enables vesicular transport of ATP across membranes[57] and NTPase, ortholog of CD39 which dephosphorylates ATP into AMP[58]. LoF of MFS18 in the MFS18^{LL00478} mutant strongly increased C4da neuron axon regeneration (Fig. 7f, g). Moreover, LoF of NTPase in the NTPase^{NP5153} and glial knockdown with two independent RNAis (repo-Gal4>4NTPase RNAiv110510 and repo-Gal4>NTPase RNAiBL62850) also increased C4da neuron axon regeneration (Fig. 7f, g). These data suggest that glial release and processing of ATP and thus adenosine constitute a major route of glial inhibition or fine tuning of axon regeneration in the PNS.

Lastly, as a first attempt to determine if the regeneration inhibition induced by adenosine-AdoR signaling may be conserved in mammals, we tested adenosine treatment on cultured rat hippocampal neurons in microfluidic chambers where the axons were allowed to regrow after aspiration (see "Methods"[38,59]). When adenosine was applied immediately after injury, we saw a trend of reduced axon regeneration at 50 μM, which became more significant at 500 μM (Fig. 7h, i). The axons also appeared to have collapsed. Adenosine treatment was equally effective when applied to the soma compartment or axon only compartment, suggesting the adenosine receptor is present throughout the neurons. This result raises the possibility that a similar glia–neuron adenosine–AdoR mechanism may be at play in mammals during axon regeneration, which warrants further investigations.

## Discussion

C4da and C3da neurons are both sensory neurons but differ in their regenerative capacities, marked by their ability to generate Ca²⁺ transients after axotomy. In this study we uncover the mechanism behind this calcium activity and show that it arises from a multi-layered interplay between neurons and glia (Fig. 7j). After axon injury, glial cells release TNF-α that binds to C4da neuron TNFRs and modulates the expression of neuronal VGCC subunits to maintain their optimal ratio. These VGCCs drive Ca²⁺ transients but rely on glial Irk1 potassium channels likely for neuronal hyperpolarization to sustain their activation. Loss of any of these components leads to failure of C4da neuron axon regeneration, while manipulation of these components can be used to promote C3da neuron axon regeneration. Conversely, glia can

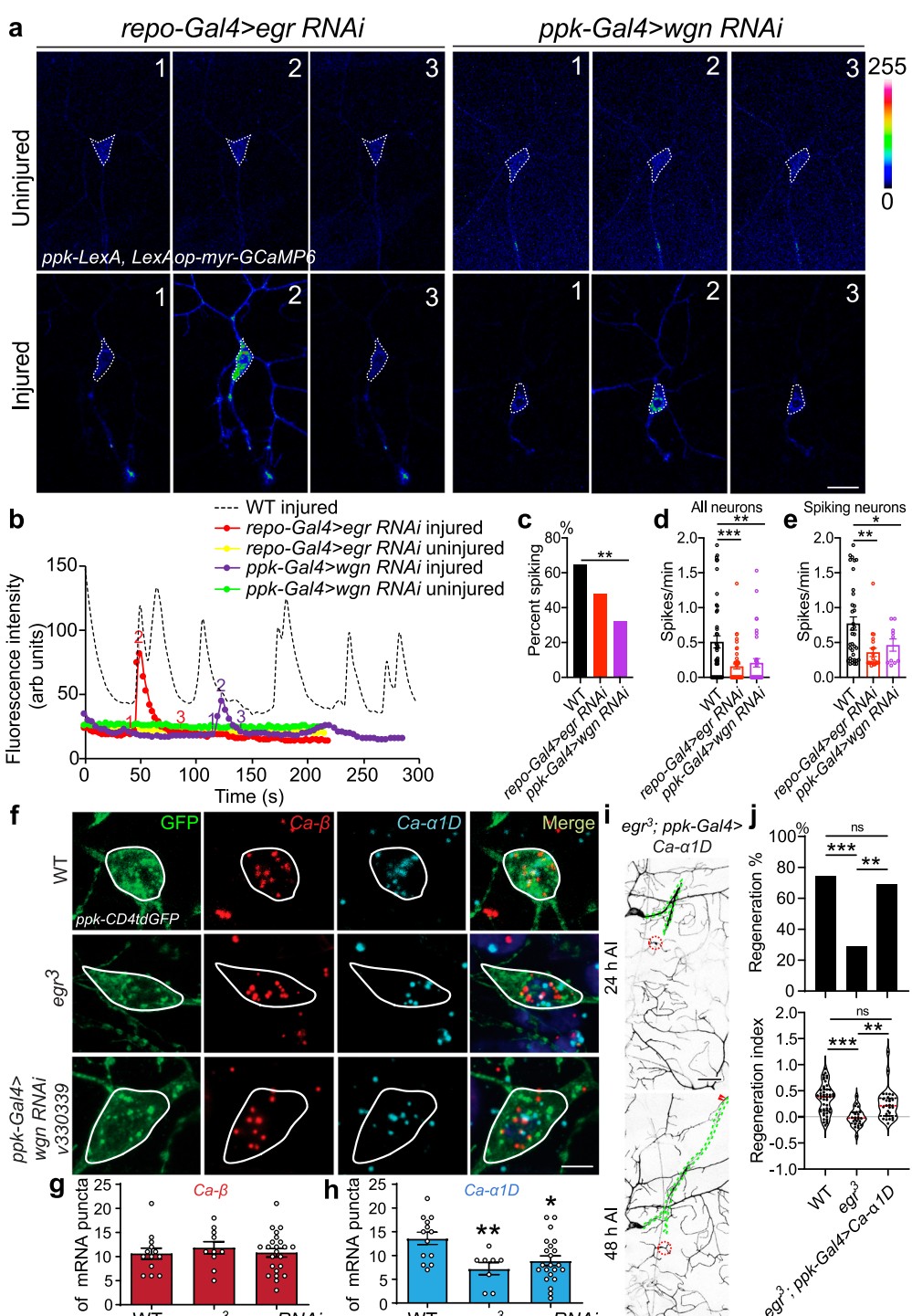

**Fig. 4 | Glia-neuron egr-wgn signaling is required for axotomy-induced Ca²⁺ transients and for maintaining *Ca-α1D* expression after injury. a** LoF of egr by glial RNAi or LoF of wgn by C4d neuron-specific RNAi reduces Ca²⁺ spikes after axonal injury. Images from three timepoints are presented. The dashed white circle marks the cell body. No obvious difference is observed in the uninjured condition. **b** Plots of the mean fluorescence intensity over time in the cell body.
**c** Quantification of the percent of neurons showing Ca²⁺ spikes. $N = 48, 46, 31$ neurons, $P = 0.1451, 0.0061$. **d**, **e** Quantification of the number of Ca²⁺ spikes per minute in all neurons (**d**, $N = 35, 48, 38$ neurons, $P = 0.0003, 0.0019$), or spiking neurons (**e**, $N = 35, 21, 10$ neurons, $P = 0.0010, 0.0442$), average values and standard error of the means are shown. **f** Expression analysis of *Ca-α1D* and *Ca-β* in injured neurons after *egr-wgn* LoF. In *egr³* mutants or C4da neuron-specific *wgn* RNAi, *Ca-β*

levels are not altered, whereas *Ca-α1D* expression is significantly reduced. The cell body is outlined with the white circle. **g**, **h** Quantification of *Ca-α1D* and *Ca-β* expression levels. $N = 13, 10, 23$ neurons, $P = 0.7776, 0.9926$ and $N = 13, 8, 22$ neurons, $P = 0.0092, 0.0138$, average values and standard error of the means are shown. **i**, **j** Overexpression of Ca-α1D in C4da neurons rescues regeneration in *egr³* mutants. The injury site is demarcated by the red dashed circle, and axons are traced by a green dashed line. Arrowheads show the regrowing axon tip. $N = 48, 24, 29$, $P = 0.6115, 0.003, 0.0058$ (top). $*P < 0.05, **P < 0.01, ***P < 0.001$, Fisher's exact test, two-sided (**c, j** top), one-way ANOVA followed by Holm−Sidak's test (**d, e**) or Tukey's test (**g, h, j** bottom). Scale bar = 20 μm (**a**) and 10 μm (**f**). Source data are provided as a Source data file.

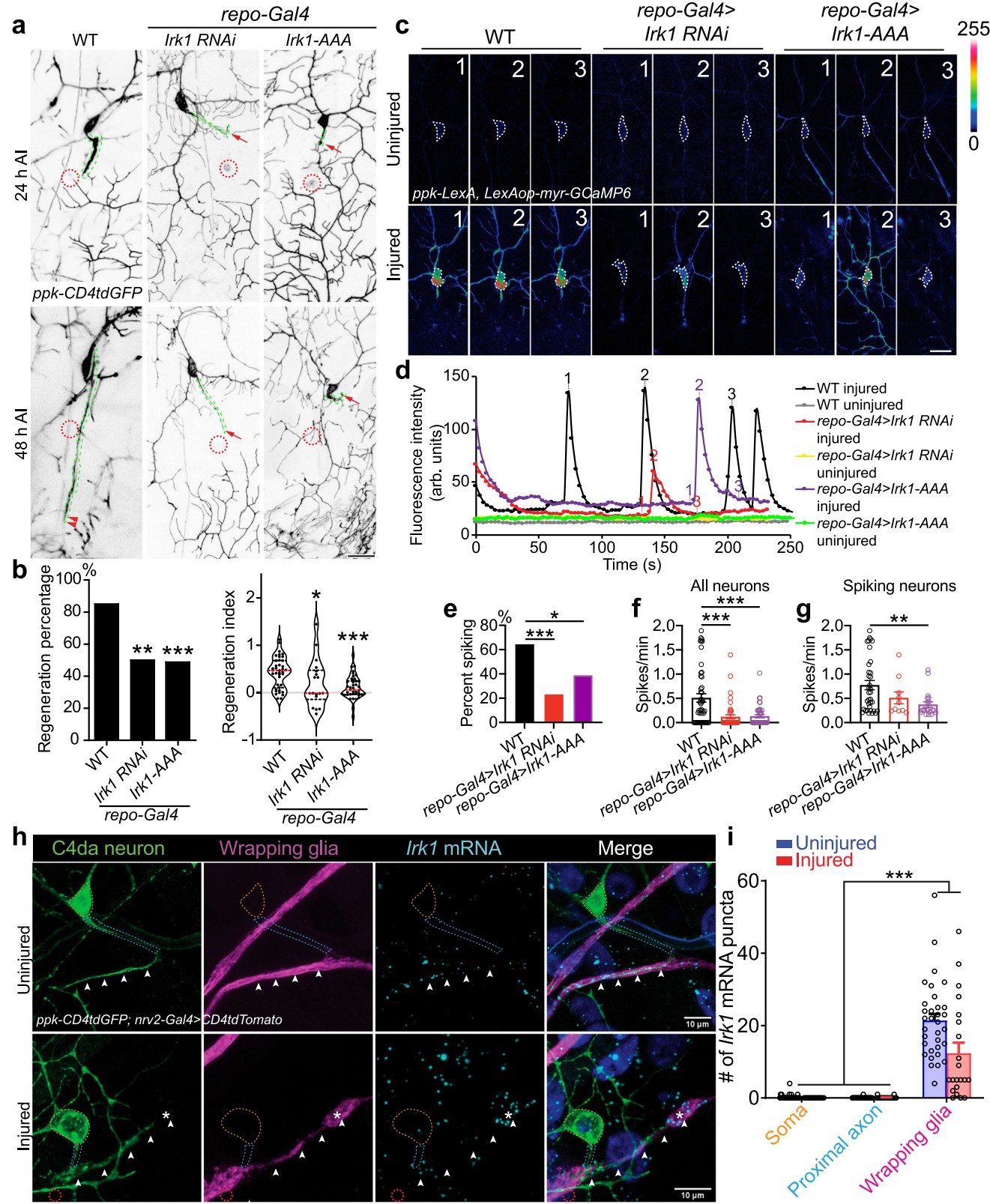

dampen regeneration by limiting neuronal hyperpolarization through AdoR-Ih signaling. Our work thus elucidates a multifaceted neuron–glia system of interaction, with glia regulating neuronal ion channel expression and glial ion channels modulating neuronal electrical activity, that controls axon regeneration. During our study, Wang and colleagues (2023) independently reported that gliotransmission and adenosine signaling promote axon regeneration[60], consistently

highlighting the importance of glial electrical signaling in instructing axon regeneration. The opposite effect of adenosine signaling observed in the two studies may be attributed to the difference in reagents or detailed methodology. We also believe that AdoR is dose sensitive and that its overexpression may in fact perturb its normal function. Thus, extra caution should be taken when interpreting results derived from the overexpression of adenosine receptors.

**Fig. 5 | *Irk1* is expressed in glial cells and is required for axon regeneration and axotomy-induced Ca²⁺ transients. a, b** LoF of glial Irk1 by RNAi or Irk1-AAA OE reduces C4da neuron axon regeneration. Arrow marks axon stalling while arrowheads show the regrowing axon tips. *N* = 40, 18, 41 neurons, *P* = 0.0087, 0.0008 (left). **c–g** Glial Irk1 is required for axotomy-induced Ca²⁺ transients in C4da neurons. (**c**) C4da neurons show Ca²⁺ spikes after axonal injury in WT. LoF of Irk1 by glial RNAi or Irk1-AAA reduces Ca²⁺ spikes. Images from three timepoints are presented. The dashed white circle marks the cell body. No obvious difference is observed in the uninjured condition. **d** Plots of the mean fluorescence intensity over time in the cell body. **e** Quantification of the percent of neurons showing Ca²⁺ spikes. *N* = 48, 43, 41 neurons, *P* = 0.0001, 0.0200. **f, g** Quantification of the number of Ca²⁺ spikes per minute in all neurons (**f**, *N* = 48, 43, 41 neurons, *P* = < 0.0001, <0.0001) or

spiking neurons (**g**, *N* = 35, 10, 20 neurons, *P* = 0.1203, 0.0062), average values and standard error of the means are shown. **h, i** *Irk1* mRNA is highly expressed in wrapping glia. *Irk1* is analyzed around the C4da neuron soma (outlined in dotted orange circle), proximal axon (dotted blue circle) and axon stretch (white arrowheads) covered by wrapping glia (*nrv2-Gal4*). With or without axon injury, *Irk1* transcripts are enriched in the wrapping glia along the axon. Axons are observed to regenerate along/towards wrapping glia with high *Irk1* expression, marked by asterisks. *N* = 21, 33 neurons, *P* = < 0.0001. average values and standard error of the means are shown. *P < 0.05, **P < 0.01, ***P < 0.001, Fisher's exact test, two-sided (**b** left and **e**), one-way ANOVA followed by Holm–Sidak's test (**b** right, **f**, **g**) or Tukey's test (**i**). Scale bar = 20 μm (**a**, **c**) and 10 μm (**h**). Source data are provided as a Source data file.

Future studies are warranted to resolve the seeming discrepancy between the two studies.

The results of our study provide insights into the temporal sequence of the regenerative process and calcium activity. Our data suggest that neurons need to express a minimum level of both Ca-α1D and Ca-β to regenerate axons, and that both the expression level and the *Ca-β/Ca-α1D* ratio are critical for the axotomy-induced Ca²⁺ transients and axon regeneration. In injured C3da neurons, the expression level of *Ca-α1D* and *Ca-β* is much lower than that of C4da neurons, which is the reason why they fail to show Ca²⁺ transients and axon regeneration, even with a proper ratio. This is further supported by the reduced Ca²⁺ transients and axon regeneration of C4da neurons when the egr-wgn axis is perturbed, which significantly reduces *Ca-α1D* expression after injury. We show overexpression of *Ca-β* or *Ca-α1D* increases STT in C3da neurons to 35% or 50%, respectively, compared to 60% resulting from double overexpression. This is in full agreement with the axon regeneration data. We depict the various scenarios resulting from manipulations of *Ca-α1D* and/or *Ca-β* in C4da and C3da neurons in the ratio hypothesis (Supplementary Fig. 3l, m), based on our interpretation of the expression and regeneration results. As our ratio hypothesis is largely based on the RNA level studies and a similar ratio of mRNA does not necessitate an equivalent amount of surface protein, more precise metrics of surface protein levels will help better understand the mechanics between channel proteins during regeneration, and more in-depth analyses in multiple systems are warranted to further substantiate the ratio hypothesis. Furthermore, the ratio hypothesis may go beyond the stoichiometry, as we expect different ratios to primarily impact ionic currents of Ca-α1D downstream of the ER exit, similarly postulated by Neely and colleagues, who hypothesized a two-state model in which a second β subunit binds after ER release and is able to modulate channel activity[61]. It is also suggested that the essential Cavβ modulatory properties are independent of the AID (α1-interacting domain in the I-II intracellular linker), as has been proposed as the primary interaction site in α1 subunits[62]. Therefore, alternatively, it is plausible that the effect we saw during regeneration is contributed by Ca-β binding to Ca-α1D to reach the second activation state. Our model, however, does not preclude other alternative possibilities.

TNF signaling allows for the proper ratio of Ca-β/Ca-α1D to be reached in C4da neurons, after which Ca²⁺ transients can initiate, supported by glial Irks, and promote axon regeneration. The spatial, temporal, and type of calcium signaling is critical for instructing downstream events, and for setting up the balance among axon degeneration, regeneration, and terminal differentiation. It has been shown that the Piezo-dependent response to mechanical forces experienced by axon growth cones may lead to local calcium influx that shuts down the regeneration program[38]. Similarly, calcium influx through N and P/Q type voltage-gated calcium channels has been reported to suppress axon regeneration in the mammalian central nervous system (CNS)[18,19,63]. Supporting our work, an L-type VGCC has also been found in *C. elegans* to be required for regeneration[64]. We thus

propose that calcium signaling is utilized at multiple phases of axon regeneration. Depending on its origin, pattern, and intensity, calcium signaling may lead to distinct downstream cascades to promote or restrict axon regeneration.

The results presented here offer insights into how axon regeneration may be promoted. Overexpression of Ca-α1D and Ca-β boosted regeneration in C3da neurons. As noted, Ca-α1D OE in C4da neurons impaired regeneration, further suggesting that it is not only the quantity, but also the ratio that is needed for optimized function, a concept which has been explored in many biological contexts—molecular titration[65]. The increased regeneration in C3da neurons occurred even though full Ca²⁺ spikes were not created in C3da neurons by the VGCC overexpression, suggesting an elevated Ca²⁺ baseline with STT may be sufficient to trigger downstream components that lead to regeneration, and the calcium signal triggering regeneration may be malleable. This was similarly seen in *C. elegans*, where fluctuating Ca²⁺ transients, rather than spikes, promoted dendrite growth[66]. The question of why WT C3da neurons do not change their Ca-β/Ca-α1D levels to promote regeneration after injury is not fully answered. The axons of C4da and C3da neurons are bundled together by the cytoplasm of the same glial cell similar to mammalian Remak bundles. After injury, our work shows that egr released by glia binds wgn receptors in C4da neurons to modulate VGCC subunit expression. However, C3da neurons do not appear to exhibit a similar pro-regeneration effect. One possibility is that C3da neurons lack sufficient wgn to respond to TNF-α. They may also lack downstream players after wgn activation that lead to expression of optimal Ca-β/Ca-α1D ratio. Another possibility is that Ca-α1D expression in injured C3da neurons may be too low compared to injured C4da neurons after injury, to reach a threshold necessary for Ca²⁺ transients and regeneration. Finally, it is also possible that despite being wrapped by the same glial cell, the glial cytoplasm may selectively release TNF-α in the outpouchings surrounding C4da axons. Our work thus implicates the relationship between neurons and glia as a possible factor controlling subtype-specific regeneration and opens further avenues of exploration.

Our study shows that TNF-α released from glia has a pro-regenerative role, by modulating neuronal L-type VGCC subunit expression. Interestingly, TNF-α is known to increase calcium entry by promoting surface expression of AMPA and NMDA receptor channels[67]. Moreover, prior work that demonstrated glial Kir4.1 channels working together with neuronal VGCCs to drive bursting also found that neuronal NMDA receptors are necessary for the bursting activity[32,33]. We thus hypothesize that glia-derived TNF-α may act to alter expression of additional factors, such as NMDA receptors in neurons to regulate regeneration.

As it was originally called *I*f to signify its "funny" nature[68], the *I*h current generated by HCN channels remains mysterious. While in our work, and in prior epilepsy research[69], HCN channel LoF confers increased Ca²⁺ spiking, the electrophysiology underlying this is enigmatic. We hypothesize that the membrane potential is clamped to the

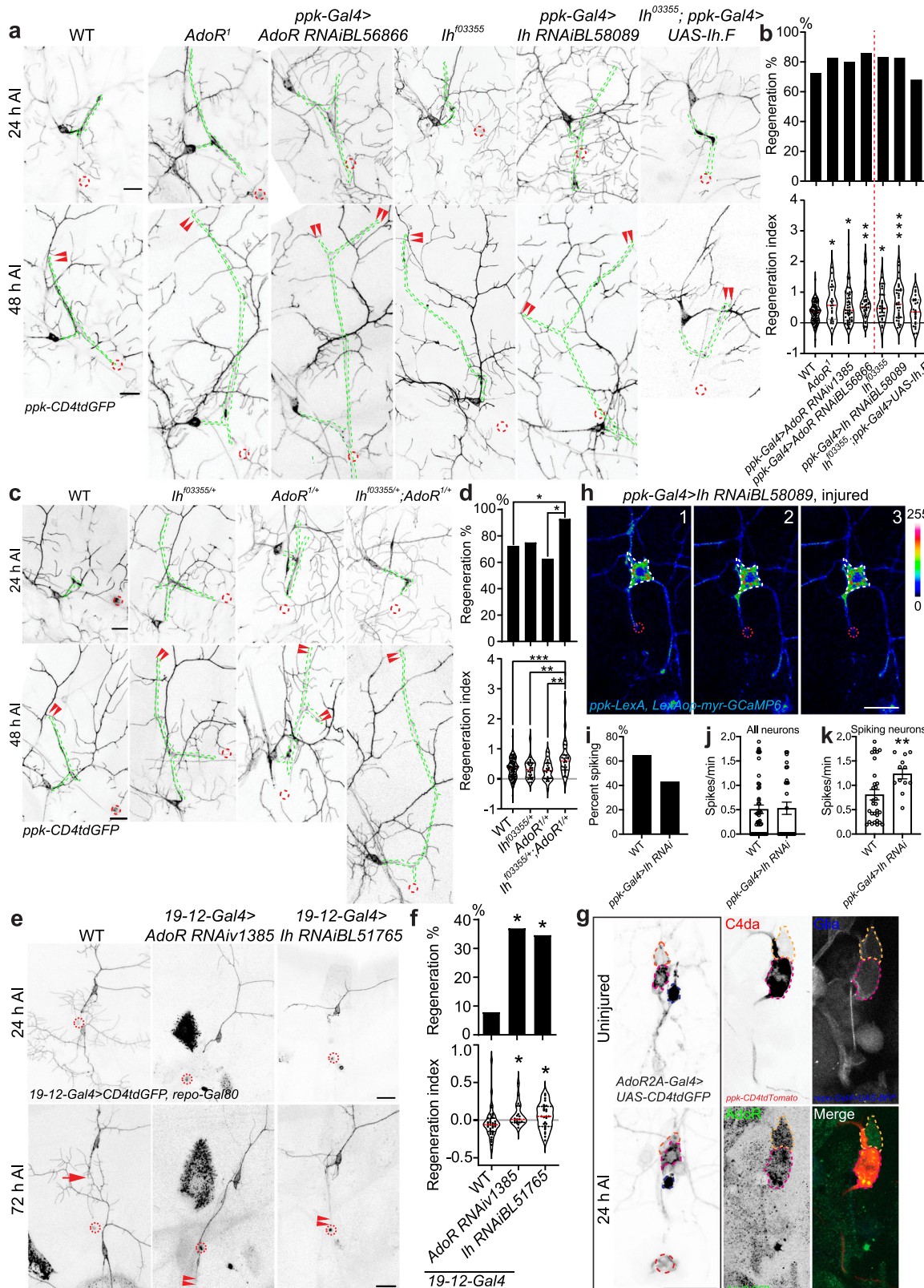

resting voltage in regions of high HCN channel density, requiring a stronger input to reach the action potential threshold, although further computational modeling is required to fully understand this mechanism. It was found that HCN channel's ability to clamp the membrane voltage prevents hyperpolarization-dependent resetting of N-type and T-type calcium channels, causing prolonged steady-state inactivation[35,69]. Other work found epileptic thalami have increased

HCN channel expression, which causes a mismatch of subunits rendering HCN channels unresponsive to cAMP, impairing their function and boosting $Ca^{2+}$ spiking[70]. Thus, similar to Ca-α1D and Ca-β, individual subunit expression after injury may play an equally important role in modulating regeneration. While our study focuses on neuron regeneration, we hope that our findings will be useful to future epilepsy research.

**Fig. 6 | The AdoR-Ih axis suppresses axon regeneration in the PNS. a** Global LoF of AdoR and Ih by mutants $AdoR^1$ and $Ih^{f03355}$, as well as knockdown by C4da neuron-specific RNAi, increases regeneration in C4da. Driving Ih expression in C4da neurons in the mutant $Ih^{f03355}$ background recovers the WT phenotype. Arrowheads show the regrowing axon tips. **b** Quantification of C4da axon regeneration with regeneration percentage and regeneration index. $N$ = 94, 23, 41, 35, 36, 22, 25 neurons. **c** Heterozygous mutations in both traits ($Ih^{f03355/+}$; $AdoR^{1/+}$), but neither trait individually, show increased axon regeneration. **d** Quantification of C4da axon regeneration with regeneration percentage and regeneration index. $N$ = 94, 28, 24, 28 neurons, $P$ = 0.0227, 0.1430, 0.0147 (top). **e** LoF in C3da of both $AdoR$ and $Ih$ showed increased axon regeneration. **f** Quantification of C3da axon regeneration with regeneration percentage and regeneration index. $N$ = 38, 19, 26 neurons, $P$ = 0.0112, 0.0101 (top). **g** Expression of AdoR with or without injury, analyzed by AdoR-Gal4 or AdoR-GFP ($AdoR^{MI01202-GFSTF.1}$). Pink and orange dashed lines mark C4da

and C3da neurons, respectively. Injury site is marked by the dashed circle. **h–k** Neuronal Ih suppresses axotomy-induced $Ca^{2+}$ transients in C4da neurons. **h** LoF of Ih by C4da neuron specific knockdown increases $Ca^{2+}$ spikes in the spiking neurons, without significantly altering the percent of neurons spiking or the spike rate in all neuron. Images from three timepoints are presented. The dashed white circle marks the cell body. **i** Quantification of the percent of neurons showing $Ca^{2+}$ spikes. $N$ = 48, 28 neurons, $P$ = 0.0932. **j, k** Quantification of the number of $Ca^{2+}$ spikes per minute in all neurons (**j**, $N$ = 48, 28 neurons, $P$ = 0.8949) or spiking neurons (**k**, 30, 12 neurons, $P$ = 0.0068), average values and standard error of the means are shown. *$P < 0.05$, **$P < 0.01$, ***$P < 0.001$, Fisher's exact test, two-sided (**b** top, **d** top, **f** top, **i**), one-way ANOVA followed by Holm–Sidak's test (**b** bottom, **d** bottom, and **f** bottom), unpaired Student's $t$ test, two-sided (**j, k**). Scale bar = 20 μm. Source data are provided as a Source data file.

The mechanism of how $Ca^{2+}$ transients lead to cytoskeletal changes that drive axon regrowth remains to be elucidated. However, the TNF-α signaling and ion channel interactions uncovered in this study provide potential targets in neurons and glia for promoting calcium activity in injured neurons. Further work in mammalian systems is warranted to determine if this TNF signaling and electrical coupling of neurons and glia takes place in dorsal root ganglion neurons or retinal ganglion cells, and whether the resultant $Ca^{2+}$ transients can promote regeneration. Intriguingly, mammalian satellite glia selectively express the Irk homolog Kir4.1, which is used as a marker for these cells in peripheral ganglia[31,71]. Furthermore, Kir4.1 levels in the trigeminal ganglion have been shown to decrease in a model of chronic nerve injury that leads to neuropathic pain[71]. Our work suggests that analysis of the electrical relationship between glia and neurons after injury in mammalian neurons could be insightful, especially investigation of the effects of TNF and Kir4.1 on calcium activity and regeneration.

Glia in the PNS have been well characterized as pro-regeneration, but our study has unexpectedly demonstrated that they can also restrict axon regeneration, through adenosine signaling. We propose that glia emit balanced pro- and anti-regeneration signals to fine-tune axon regeneration, allowing them to readily adapt to environmental cues. This could lend insights as to why glia act so differently in the PNS versus CNS. The findings of this work also show a host of potential therapeutic targets for both sparking the axon regeneration of otherwise static neurons and enhancing the regenerative abilities of neurons. Through this investigation, we identified the complex control of neuron $Ca^{2+}$ transients after injury, which may have implications beyond neuron regeneration in both mammalian depression and epilepsy through mammalian Kir4.1 and HCNs, respectively. The factors triggering glial TNF-α and ATP release after injury remain unknown. Similarly, future investigations unveiling the factors downstream of these $Ca^{2+}$ transients will be instrumental in our understanding of axon regeneration. As a whole, the findings presented provide both therapeutic targets and extensive framework for further investigation.

## Methods
### Fly stocks
$UAS$-$Kir2.1$[40], $19$-$12$-$Gal4$[72], $repo$-$Gal80$[73], $ppk$-$CD4$-$tdGFP$[74], $Ca$-$β^{CR01SS4-TG4.1}$ [75], $UAS$-$RedStinger$[76], $ppk$-$Gal4$[74], $nompC$-$Gal4$[77], $GCaMP6(s)$[37], $ppk$-$LexA$[78], $NompC$-$LexA$[79], $LexAop$-$myr$-$GCaMP6(s)$[38], $repo$-$Gal4$[80], $UAS$-$Irk$-$WT$ and $UAS$-$Irk1$-$AAA$[48], $egr^3$ [44], $egr$-$Gal4$, $UAS$-$wgn.Venus$[43], $wgn^{22}$ [45], $wgn^{fTRG00673.sfGFP-TVPTBF}$ [47], $AdoR^1$ [49], $AdoR^{MI01202-GFSTF.1}$ [51,52], $AdoR^{2A-GAL4.KI}$ [53], $Ih^{f03355}$ [50] and $UAS$-$iATPSnFR$[56] have been previously described. $UAS$-$Ca$-$α1D$ $RNAiBL33413$, $UAS$-$Ca$-$α1D$ $RNAiBL25830$, $Df(2L)Exel6028$, $UAS$-$Ca$-$β$ $RNAiBL43292$, $UAS$-$Ca$-$β$ $RNAiBL29575$, $Df(2L)BSC293$, $UAS$-$Ca$-$α1T$ $RNAiBL39209$, $UAS$-$Ca$-$α1T$ $RNAiBL26251$, $UAS$-$Irk1$ $RNAiBL25823$, $Traf6^{G904}$, $UAS$-$AdoR$ $RNAiBL56866$, $UAS$-$Ih$ $RNAiBL51765$, $UAS$-$Ih$ $RNAiBL58089$, $UAS$-$NTPase$ $RNAiv110510$, $UAS$-$NTPase$ $RNAiBL62850$ were from the Bloomington stock center. $UAS$-$Ca$-$β$ $RNAiv102188$, $UAS$-$egr$

$RNAiv45253$, $UAS$-$egr$ $RNAiv330339$, $UAS$-$wgn$ $RNAiv9152$, $UAS$-$Irk1$ $RNAiv107389$, $UAS$-$AdoR$ $RNAiv1385$, were from VDRC. $NTPase^{NP5153}$ and $MFS18^{PBac\{SAstopDsRed\}LL00478}$ ($MFS18^{LL00478}$) were from Kyoto Stock Center. To generate the $UAS$-$Ca$-$α1D$, $UAS$-$Ca$-$β$ and $UAS$-$Ih.F$ stocks, the entire coding sequences were cloned into the pACU2 vector, and the constructs were then injected (Rainbow Transgenic Flies, Inc). Randomly selected male and female larvae were used. Analyses were not performed blind to the conditions of the experiments. In our study, we typically used one mutant plus one RNAi knockdown, or two independent RNAi strains to confirm the phenotype of each candidate gene. No difference is observed in axon regeneration between male and female flies, and both genders were used. All studies and procedures involving animal subjects were performed under the approval of the Institutional Biosafety Committee (IBC) at the Children's Hospital of Philadelphia.

### Sensory axon injury and quantification
Axon injury and confocal imaging were performed according to published methods[8,13,81]. Briefly, larvae at -72 h after egg laying (h AEL) were anesthetized with diethyl ether and 3−7 neurons per larvae were injured via a focused 930 nm two photon laser (-500 mW). Injured larvae were placed on grape agar and yeast plates at room temperature to recover, and again anesthetized and imaged live at appropriate stages. Quantification of axon regeneration was performed according to published methods[8,13]. Briefly, regeneration percentage depicts the percent of regenerating axons among all the axons that were lesioned. Regeneration index was calculated by normalizing the length of regenerated axon to the distance between the cell body and the axon converging point (DCAC). An axon is defined as regenerating only when it obviously regrew beyond the retracted axon stem. The regeneration parameters from various genotypes were compared to that of the WT if not noted otherwise, and only those with a significant difference were labeled with the asterisks. The direction of axon regeneration was quantified either as a percentage, representing the proportion of neurons which extended in each direction, or as a ratio. The ratio was quantified according to published methods, which briefly was calculated as the ventral growth minus the dorsal growth over the total growth of the axon[54].

### RNAscope
RNAscope (ACDbio) was used to image $Ca$-$α1D$, $Ca$-$β$ and $Irk1$ mRNAs, according to the manufacturer's instructions. $Drosophila$ larval body walls were fixed, dehydrated, and exposed to probes targeting $Ca$-$α1D$, $Ca$-$β$, and $Irk1$. Subsequent amplification steps were used to amplify the fluorescent dye to allow for single mRNA imaging. Probes targeting $bacteria mRNA$ were used as a negative control and exhibited no staining normal background. The mRNA for $GFP$ was used as our positive control and was only expressed in the appropriate class of neurons with expected ectopic GFP expression. Fluorescent puncta

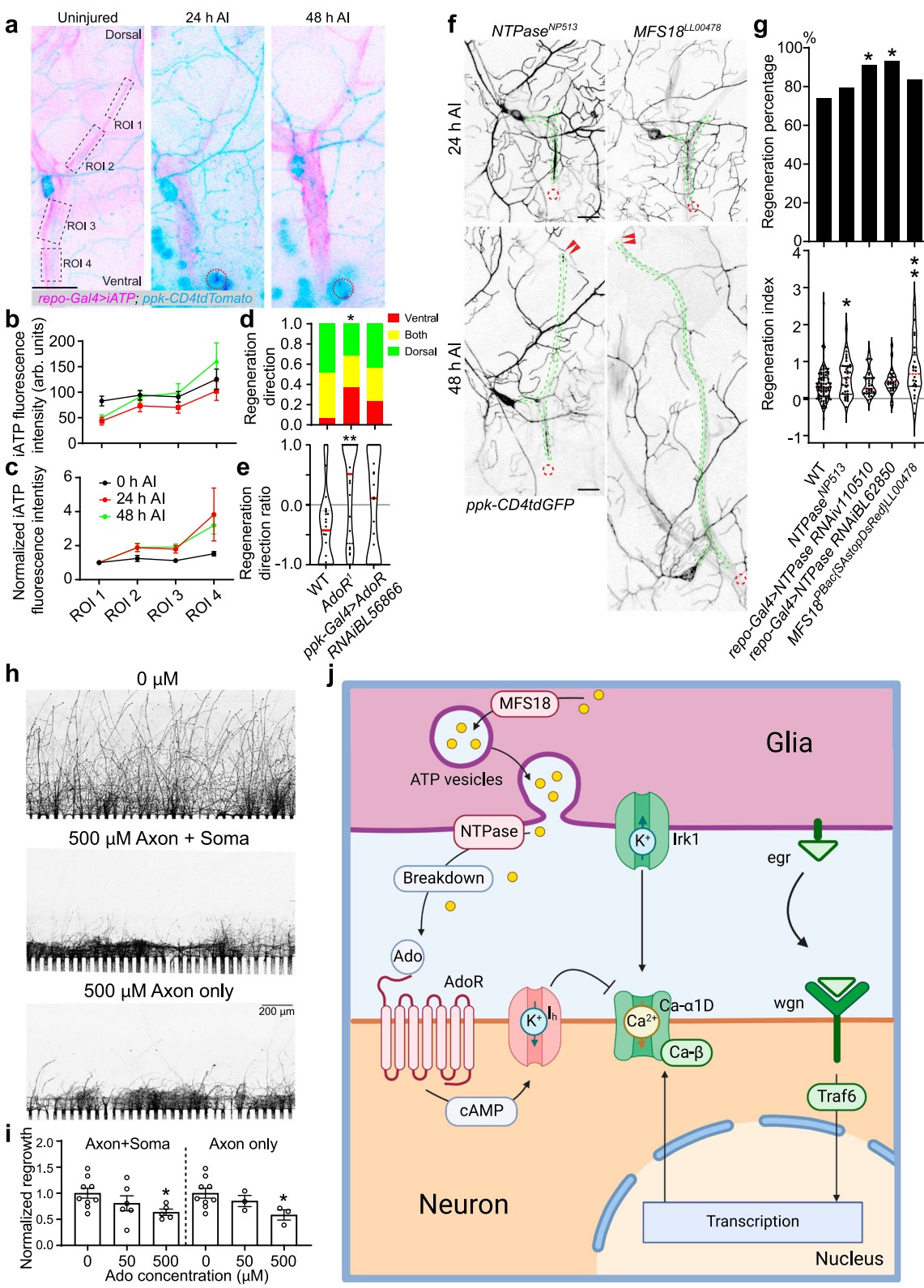

represent individual mRNA, whereas fluorescent aggregates are comprised of multiple mRNAs. Following the RNAscope protocol, immunolabeling for GFP (chicken anti-GFP, ab13970, 1:100, Abcam; fluorescence-conjugated secondary antibodies, 1:1000, Jackson ImmunoResearch) was performed. GFP-positive cells were included for analysis. Individual mRNA puncta were imaged using a confocal microscope and the average number of puncta per neuron was quantified using z-stack images in ImageJ.

## Generation of antibodies

Anti-Ca-α1D II, anti-Ca-β, and anti-Irk1 polyclonal antibodies were generated by injecting specific antigens into rabbits (Abmart, Shanghai). The animal is immunized repeatedly to obtain higher titers of antibodies specific for the antigen. Antigens for Ca-α1D and Irk1 antibodies were peptides (Ca-α1D, KSKDTSQIAESDIVEG; Irk1, DLVA-NITHTVSVTQVSN, respectively, with an internal cysteine replaced by a serine). The Ca-α1D antigen is predicted to be on the cell surface and

**Fig. 7 | Glial release of ATP/Ado suppresses axon regeneration. a** Glial ATP was monitored with *repo-Gal4 > iATPSnFR* and is shown in pink. The four regions of interest (ROI) are marked by the dashed black squares. Greater iATP fluorescence can be seen in the ROIs ventral to the cell body at 24 and 48 h AI, demonstrating an ATP gradient. The injury site is marked by the dashed red circle. **b, c** Plots of the fluorescence intensity over the four regions, with (**c**) being normalized to the fluorescence of ROI 1. $N = 8$ neurons per ROI per timepoint, average values and standard error of the means are shown. **d** The plot of the regeneration direction with regard to proportion of regenerating axons. AdoR LoF with mutant or RNAi shows a greater proportion of ventral-only regenerating axons. $N = 24, 20, 22$ neurons, $P = 0.0184, 0.0807$. **e** A plot of the regeneration ratio (Methods) shows AdoR LoF increases the regeneration ratio, indicative of more ventral growth than WT. $N = 24, 20, 22$ neurons, $P = 0.0084, 0.1688$. **f** Global LoF of NTPase by mutant $NTPase^{NP513}$ increases regeneration in C4da. Glial knockdown of NTPase using two RNAis does not significantly increase the regeneration index but instead boosts the percentage of regenerating axons. Global LoF of MFS18 by mutant $MFS18^{LL00478}$ greatly enhances C4da axon regeneration. Arrowheads show the regrowing axon tips. **g** Quantification of C4da axon regeneration with regeneration percentage and regeneration index. $N = 84, 39, 33, 29, 24$ neurons, $P = 0.6524, 0.0473, 0.0344, 0.4241$ (top). **h** Adenosine treatment reduces axon regeneration of rat hippocampal neurons cultured in a microfluidic chamber when applied either to the whole neuron or only the axon. The axons are labeled with CTB488. **i** The axon coverage area is measured and normalized to the total width of the microgrooves. $N = 9, 6, 5$ microfluidic chambers, $P = 0.3185, 0.0240$, and $N = 9, 3, 3$ microfluidic chambers, $P = 0.9578, 0.0460$ m, average values and standard error of the means are shown. **j** The proposed glia–neuron coupling of axon regeneration. The illustration in (**j**) was created using Biorender.com. $*P < 0.05, **P < 0.01$, Fisher's exact test, two-sided (**d, g** top), one-way ANOVA followed by Holm–Sidak's test (**e, g** bottom), by Dunn's test (**i**). Scale bar = 20 μm (**a, f**) and 200 μm (**h**). Source data are provided as a Source data file.

---

can be detected using a nonpermeable staining condition. For the Ca-β antibody, part of Ca-β (300aa beginning from DREFDWDQGGTSYQ) was expressed as antigen. The specificity of these antibodies was confirmed by the reduced labeling in fly larvae after RNAi knockdown (Supplementary Figs. 3c–f and 6e, f).

## Immunohistochemistry

Third instar larvae were fixed and stained according to standard protocols. Briefly, larvae fillet preps were fixed in 4% paraformaldehyde/PBS for 20 min at room temperature, and stained with the proper primary antibodies and subsequent secondary antibodies, each for 2 h at room temperature. The following antibodies were used: chicken anti-GFP (ab13970, 1:100, Abcam), goat anti-HRP (123-605-021, 1:200, Jackson ImmunoResearch), rabbit anti-RFP (600-401-379, 1:500, Rockland Immunochemicals), rabbit anti-Ca-α1D (1:50)[41], rabbit anti-Ca-α1D II (1:100, this study), rabbit anti-Ca-β (1:100, this study), rabbit anti-Irk1 (1:100, this study), and fluorescence-conjugated secondary antibodies (1:1000, Jackson ImmunoResearch).

## In vivo calcium imaging and quantifications

For calcium imaging, larvae were not anesthetized with diethyl ether, as it may silence neuronal activity. Instead, uninjured larvae or injured larvae at 24 h AI were immobilized in a small groove indented into Modeling Clay and covered with a glass coverslip, held in place with vacuum grease on the microscope slide. Each neuron was imaged via time-lapse using a Z-stack (4 microns depth) focusing on the soma for a total of 80 cycles. Given the lack of anesthesia, larva movement occasionally required resetting and capturing multiple time-lapse videos for some neurons for a total of ~5 min of data per neuron. Calcium imaging videos were analyzed by manually scoring the number of spikes. A spike was defined as a visually recognizable and intense increase in fluorescence intensity throughout soma and dendrites. Spikes were also assessed quantitatively in terms of a fluorescence threshold. In order to be assessed as a spike, the ratio of the highest mean fluorescence of the soma during the spike in question to the average fluorescence of the soma at rest needed to meet or exceed 1.608. Average intensity values were calculated by measuring the mean gray value of the soma in ImageJ. The threshold of 1.608 was selected as it is 2 standard deviations above the average ratio of the maximum mean soma fluorescence during a spike to mean fluorescence of the resting soma in all WT C4da spiking neurons. We defined the soma to be at rest when the fluorescence in the soma was not visually changing (i.e. when the neuron was not actively spiking). Any time points with fluctuations in fluorescence were excluded and the remaining time points were averaged to calculate the mean resting fluorescence. For C3da neurons, uninjured neurons showed $Ca^{2+}$ spikes correlated with visible movement of the larva. Since these neurons sense gentle touch stimuli that could occur during movement against the clay and coverslip, we only counted neurons as spiking if spikes occurred without larval movement when scoring. The percent of neurons showing spiking was calculated, as well as the average spiking rate (total number of spikes/total time imaged for a given neuron) for all neurons (both spiking and non-spiking) and spiking only. Example traces of fluorescence intensity in figures were generated by measuring mean gray value of the soma in ImageJ. To further analyze $Ca^{2+}$ transients in C3da neurons, we also assessed JAS and STT. JAS are defined as spikes that occur while the larva is moving (but soma displacement <20 μm). STT were defined as weaker but visually-recognizable increases in fluorescence that did not cross the 1.608 threshold. Typically, ratios of the maximum average soma fluorescence during the STT to the average soma at rest fell between 1.25 and 1.60. We further compared the average resting fluorescence of WT C3da neurons to that of Ca-α1D + Caβ overexpression to assess the baseline increase of calcium. Neurons that had too much movement to calculate a reliable baseline were excluded.

## Adenosine treatment

Microfluidics devices were produced as previously described[38]. Chip design: the microfluidic device was made up of two elements: macrochambers (length 7000 μm, width 1200 μm, height 250 μm) for cell or fluid injection, separated by narrowing arrays micro-channels (length 450 μm, width 10 μm, height 3 μm) allowing directional axonal outgrowth. Microfluidic chip production: Polydimethylsiloxane (PDMS, Sylgard 184, Dow Corning, USA) was mixed thoroughly with a curing agent (9:1 ratio). The resulting preparation was poured onto the master and degassed in a vacuum desiccator for 3 h. The polyester resin replicate was cured at 80 °C for 6 h. The elastomeric polymer print was detached and reservoirs were punched for each macro-channel. Microfluidics culture platform fabrication: the PDMS pieces were sterilized with 70% ethanol and exposed to UV (254 nm) for 10 min. All following work performed with microfluidics devices was done aseptically in a biosafety cabinet. Once dried, the PDMS pieces were gently pressed on the cover glass (Fisher Scientific Premium Cover Glass sized: 24 × 60-1 mm) until a complete seal formed and no bubbles remained between the PDMS molds and glass surface. The cover glass had been previously coated in 0.1 mg/mL poly-D-lysine for 48 h, washed 5 times in sterile double distilled water, and thoroughly dried via aspiration.

Each microchannel of each microfluidics device was pre-filled with standard neuron medium (Neurobasal medium supplemented with B-27 at 1× working concentration, and Glutamax at 1× working concentration). Next, embryonic primary rat hippocampal neurons (E18) were obtained from Neuron R Us (University of Pennsylvania, Penn Medicine Translational Neuroscience Center) and plated at 100,000 cells/3 μL into each "soma" chamber. The neurons were allowed to adhere to the substrate for 10 min before filling each reservoir of the chambers with approximately 300 μL of standard neuron. Media changes were performed on all devices every 3 days. The neurons were

cultured for 7 days in vitro (DIV7) before being exposed to adenosine and axotomized. Adenosine was dissolved in neuron growth medium to make a stock concentration of 7 mg/mL and stored at 4 °C for up to 7 days.

The working concentrations of adenosine were used at 50 and 500 μM by diluting the stock adenosine in neuron growth medium. Axotomy was achieved via aspiration through the axon chamber to induce axon "shearing", and followed by three washes with sterile double distilled water to clear away axon debris. Adenosine solutions were applied to either the axon chamber only, or the soma and axon chambers simultaneously at 0 h after axotomy, and axons were allowed to regenerate for 24 h. To trace the axons, at 20 h after injury, Alexa Fluor 488 cholera toxin B subunit (CTB488) (Thermofisher C34775) diluted in high potassium buffer[82] to a working concentration of (5 μg/mL) was administered to the axon chamber. After 30 min, CTB488 was replaced with the appropriate growth medium and cells were incubated at 37 °C and 5% $CO_2$ for 4 h to stain the axons and cell bodies. Thus, the cells were given a total of 24 h to regenerate.

The growth media of each chamber was quickly and gently replaced by a fixative solution of 4% sucrose/4% paraformaldehyde (PFA). Each PDMS piece was quickly and gently removed, thus leaving behind cells still attached to the cover glass, and the excess fixative solution was carefully removed. A hydrophobic boundary was drawn around the cell perimeter and an additional 500 μL of fixative solution was added to each cell population. Fixation occurred at room temperature for 10 min and cells were washed three times in 1× PBS for 10 min each, without agitation. Cover glass was mounted onto microscope slides (Fisher Scientific: 125447) containing 5 μL of mounting medium (VECTASHIELD Antifade Mounting Medium: H-1200-10). Each microscope slide was imaged on a Zeiss CellDiscoverer 7 confocal microscope and regeneration was analyzed in ImageJ as described previously[38].

### Image acquisition and quantification

Images were acquired with a Zeiss LSM880 or CD7 confocal laser scanning microscope with Zen 2012 software, or Leica SP8 confocal microscope with LAS X software. The obtained images were processed and analyzed using ImageJ 1.52q.

### Statistics and reproducibility

No statistical methods were used to pre-determine sample sizes but our sample sizes are similar to those reported in previous publications[8,13], and the statistical analyses were done afterward without interim data analysis. Data distribution was assumed to be normal but this was not formally tested. Each experiment was repeated independently at least three times. Measurements were taken from distinct samples. The *N* values (sample size) are provided in the figure legends. Data are expressed as mean in bar graphs for percentage, mean ± SEM in bar graphs and scatter plots. No data points were excluded except mentioned otherwise. Two-tailed unpaired Student's t-test was performed for comparison between two groups of samples. One-way ANOVA followed by multiple comparison test was performed for comparisons among three or more groups of samples. Two-sided Fisher's exact test was used to compare the percentage. Statistical significance was assigned, $*P < 0.05$, $**P < 0.01$, $***P < 0.001$. Statistical analyses were performed using GraphPad Prism 8.

### Reporting summary

Further information on research design is available in the Nature Portfolio Reporting Summary linked to this article.

## Data availability

All data supporting the findings of this study are provided within the paper and its Supplementary Information. Correspondence and requests for materials should be addressed to Y.S. (songy2@chop.edu). Source data are provided with this paper.

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

## Acknowledgements

We thank E. Bates, G. Davis, P. Baker, M. Zurovec, and T.R. Clandinin for fly lines; J. Kuwada for the Ca-α1D antibody; Bloomington Stock Center, VDRC and Kyoto Stock Center for fly stocks; GenScript for plasmids; Neuron R Us (University of Pennsylvania, Penn Medicine Translational Neuroscience Center) for providing cultured rat neurons; and members of the Song lab for helpful discussions. This work was supported by NIH grants (1R01NS107392, 1R01NS126541) to Y.S., a Pennsylvania Department of Health grant (4100088540) to Y.S., an NIH Institutional Research and Academic Career Development Award (IRACDA) postdoctoral fellowship and an NIH Ruth L. Kirschstein National Research Service Award (NRSA) Institutional Research Training Grant in Neurodevelopmental Disabilities (T32) to A.M., a Ministry of Science and Technology China Brain Initiative grant (STI2030-Major Projects 2022ZD0204700) to F.L., Shanghai Rising-Star Program (22QA1402200) to F.L., and a National Natural Science Foundation of China grant (No. 32271188) to F.L.

## Author contributions

Experimental design, J.P., P.G., F.L. and Y.S.; methodology, S.T., J.P., P.G., A.M., L.M., F.L. and Y.S.; data collection and analysis, S.T., J.P., P.G., A.M., X. L., T.O., T.S., L.M., Q.W., S.W., J.Q., Q.L., F.L. and Y.S.; writing—original draft, S.T., J.P., P.G. and Y.S.; writing—review and editing, T.S., A.M., and F.L.; funding acquisition, Y.S.; supervision, Y.S.

## Competing interests

The authors declare no competing interests.
