## [Peer Review File · Nature Communications]

Glia instruct axon regeneration via a ternary modulation of neuronal calcium channels in *Drosophila*Reviewers' Comments:

Reviewer #1 (Remarks to the Author):

The manuscript submitted by Li and colleagues to Nature Communications describes the interaction between neurons and glial cells in the regulation of intracellular axonal Ca²⁺ to promote nerve regeneration. The authors describe how the Ca²⁺ channel Ca- α 1D and its subunit Ca- β can positively mediate the response to axonal injury. The authors then proceed to describe the role of glial cells and their expression of the IrK1 potassium channel on axonal Ca²⁺. Finally, the effects of TNF on the levels of Ca- α 1D and the promotion of regeneration is explored. The finding that the glia can control neural activity after injury to regulate regeneration is interesting and adds an additional angle from which to look at neuronal regeneration and its possible therapeutic approaches. While the manuscript uses an extensive arrays of genetic manipulations and provides a series of compelling data, the flow of the study itself feels disjointed. There are three different lines of investigations, which should be better related to each other. Specifically, while there are logical connections established between Ca- α 1D/Ca- β and IrK1 as well as between TNF signaling and Ca- α 1D/Ca- β , the link between IrK1 and TNF signaling is not sufficiently fleshed out. In addition, the link between TNF signaling and Ca- α 1D/Ca- β feels underdeveloped and would benefit from further characterization. The authors might consider to simplify the manuscript, by strengthening the relationship between Ca- α 1D/Ca- β and IrK1, while further developing the TNF part of the study in a separate manuscript. In addition, the difference between C4da and C3da neurons is addressed in regards to their different response to Ca- α 1D and Ca- β expression and the generated Ca²⁺ spikes. It is unclear why there is a boost in regeneration for both cell types, but their spikes are very different (which is crucial as Ca²⁺ spikes are central to the message of the paper). Though the authors discuss this aspect, it should be further developed. This important part of the study is also dropped in favor of other lines of investigation. The difference in behavior of C4da and C3da neurons should be further investigated and explained.

There are a few minor points, which should be addressed:

1. Throughout the manuscript there are inconsistencies regarding the order of appearance of the figure panels. For examples Figure 2h and 2i are mentioned in the text before figure 2c, g and j. A part of Figure 5a-c is discussed only at the end of the result sections, well after Figure 7 is introduced. The authors might want to rethink the figures, so that these inconsistencies are removed.
2. In Figure 6c there is no WT reference

Reviewer #2 (Remarks to the Author):

In the manuscript "Coding axon regeneration by a glia-neuron ion channel module and TNF signaling" the authors follow up on previous work suggesting that neuronal calcium and glia control axon regeneration potential in Drosophila sensory neurons. In this manuscript they focus on a role for voltage-gated calcium channels (specifically Ca- α 1D and Ca-beta) in neurons and an inward rectifying potassium channel in glia. They also show that TNF signaling between neurons and glia has some impact on axon regeneration. While there are some intriguing observations, the model has critical holes, and important controls are missing. It is therefore difficult to know whether the model they present in their last figure will hold up when these holes and controls are addressed.

The model they present in the abstract, title and final figure is that glial wrapping around the axon modulates Ca availability for entry through VGCCs through potassium buffering in glia. This model assumes that the VGCC is localized to the part of the cell that is wrapped by glia. They do not provide localization data for Ca-a1D, and the functional data from this study and other work suggests that it is abundant in dendrites, which are not wrapped by glia. In this work they show global calcium spikes throughout neurons, including in dendrites suggesting widespread distribution of Ca-a1D. A substantial dendritic population of Ca-a1D is supported by previous work showing Ca-a1D and Ca-beta function in dendrites after they have been disconnected from the cell body in pruning (1). The higher levels of Ca-a1D in neurons with large dendrites (Figure 1B) is also consistent with substantial presence in dendrites. Much of the Class IV dendrite arbor is hundreds of microns away from any glial wrap, as seen in Figure 3A, right column.

The manuscript also focuses extensively on the role of Ca spikes in axon regeneration, which they show are not important for improved regeneration in Class III neurons that express Ca-a1D and Ca-beta in Figure 2. It is therefore difficult to know whether focusing on spikes in the other figures is central to the biology in question.

Critical missing controls

1. It is now standard and expected that antibody staining experiments include specificity controls. It seems appropriate to do the same for the RNAscope used extensively in multiple figures. The parallel controls to those expected for antibodies would be using the same RNA probes in knockdown cells or mutant animals. For example, for Figure 1 it would be good to knock down Ca-a1D and Ca-beta in neurons and show that the spots they quantitate go away. For figure 4, it is concluded that Irk1 spots are in glia- but the spots are on a bundle of axons and glia. A control of glial knockdown of Irk1 would strengthen this conclusion considerably.

2. In Figure 3 regeneration is compared with and without glial injury. Presumably cutting glia+axons involves considerably more tissue damage than cutting axons while leaving surrounding glia intact. The authors do not control for the added tissue damage in the glial injury condition so it is very difficult to interpret here (or in the paper they refer to) whether the key difference is the glial damage or amount of damage. In addition, the glial injury is not described in the Methods.

3. The effect of various knockdowns on cell shape before injury is not assayed, and so it is difficult to know whether the effect on regeneration is due to some large scale change in cell architecture (of neurons or glia) before injury, or whether it is a specific injury response defect. For example, in figure 3F the images for Irk1 knockdown in glia look like the neurons have increased branching near the cell body. Is neuron shape altered at baseline in Irk1 knockdown? Do glia still contact axons normally in Irk1 knockdown? Similarly does egr or wgn knockdown affect neuronal or glia development or do they function specifically after injury as suggested?

Additional points

It looks like there is data for two different types of class IV neurons in Figure 1F-I, although I could not find this specified anywhere. In the figure the blue bars are ddaC, but I am guessing that the data in 4I must be for a different class IV neuron as the regeneration index is quite different. It is not mentioned in the text why these two class IV neurons would have such different regeneration indices.

An alternate interpretation for the membrane clumps in glia in S2: when the axon is cut, the glia that wrap them are also damaged- and these blobs are part of a repair pathway.

In Figure 4 it looks like there is substantial damage to the glia in the images in A even though this is supposed to be an experiment where glia are not damaged. This raises a problem for quantitation of RNA in the wrapping glia. It looks like the area of the glial wrap is substantially reduced so the reduction in puncta shown in B is likely just due to reduction of glial area to examine.

In figure 5D why is Rab5 used as a transcription reporter driven by wgn-Gal4? Is it possible that endocytosis or something else affects Rab5 stability after injury?

Figure 7 is somewhat confusing- the condition that rescues regeneration in Figure 5C (*egr3*, UAS-Ca-a1D) has comparable levels of Ca1D to the neighboring condition that does not regenerate (*wgn* RNAi). How does this fit with the model that Ca-a1D levels are a critical regulator of regeneration.

1. Kanamori T, Kanai MI, Dairyo Y, Yasunaga KI, Morikawa RK, Emoto K. Compartmentalized Calcium Transients Trigger Dendrite Pruning in *Drosophila* Sensory Neurons. *Science*. 2013. Epub 2013/06/01. doi: 10.1126/science.1234879. PubMed PMID: 23722427.

Reviewer #1:

The manuscript submitted by Li and colleagues to Nature Communications describes the interaction between neurons and glial cells in the regulation of intracellular axonal Ca^{2+} to promote nerve regeneration. The authors describe how the Ca^{2+} channel Ca- α 1D and its subunit Ca- β can positively mediate the response to axonal injury. The authors then proceed to describe the role of glial cells and their expression of the Irk1 potassium channel on axonal Ca^{2+} . Finally, the effects of TNF on the levels of Ca- α 1D and the promotion of regeneration is explored.

The finding that the glia can control neural activity after injury to regulate regeneration is interesting and adds an additional angle from which to look at neuronal regeneration and its possible therapeutic approaches. While the manuscript uses an extensive arrays of genetic manipulations and provides a series of compelling data, the flow of the study itself feels disjointed. We appreciate the feedbacks. We have significantly revised the manuscript and further developed our story. We have now included the identification of the neuronal core machinery dictating regeneration cell type specificity – the L-type calcium channels, and revealed three pathways through which glia regulate axon regeneration, all of which impinge onto the neuronal L-type calcium channels. We hope the Reviewer will find our story tightened and strengthened. I have attached a summary of the findings below of our completely revamped story, for reference.

1. We established an *in vivo* calcium imaging paradigm in fly larvae without anesthesia, which allowed us to reveal the axotomy-induced Ca^{2+} transients only in regenerative neurons.
2. The axotomy-induced Ca^{2+} transients are mediated by L-type calcium channels, constituted by the pore-forming subunit Ca- α 1D and its modulating subunit Ca- β . They are required for axon regeneration, and exhibit differential baseline expression, response to injury, and relative ratio in regenerative versus non-regenerative neurons. NEW DATA.
3. The Ca- β /Ca- α 1D ratio correlates with regeneration cell-type specificity. Co-expression of Ca- α 1D and Ca- β confers regenerative potential to regeneration-incompetent neurons by increasing Ca^{2+} transients.
4. Peripheral glia are required for the axotomy-induced Ca^{2+} transients and hence axon regeneration.
5. Glia-derived TNF (*egr*) acts through its neuronal TNF receptor (*wgn*) to counteract the injury-induced Ca- α 1D reduction, to maintain Ca^{2+} transients and facilitate regeneration.
6. Glial inwardly-rectifying potassium channels (*Irk1*) are enriched around the axons, hypothesized to buffer K^+ ions and enhance membrane hyperpolarization, which is required for maintaining calcium channel opening. Glia-specific gain and loss of *Irk1* bidirectionally regulates neuronal Ca^{2+} transients and axon regeneration.
7. Glia also release ATP, which is converted into adenosine (Ado) and act through neuronal adenosine receptors (AdoR) to activate the hyperpolarization activated cyclic nucleotide gated potassium and sodium channel (HCN) – Ih. Ih functions as a selective low threshold filter to dampen Ca^{2+} transients and thus inhibit axon regeneration. NEW DATA.
8. Inhibiting glial ATP release, ATP to Ado conversion, neuronal AdoR and Ih all lead to enhanced axon regeneration. NEW DATA.
9. Ado treatment in cultured mammalian neurons in microfluidic chambers also impairs axon regeneration. NEW DATA.

There are three different lines of investigations, which should be better related to each other. Specifically, while there are logical connections established between Ca- α 1D/Ca- β and *Irk1* as well as between TNF signaling and Ca- α 1D/Ca- β , the link between *Irk1* and TNF signaling is not sufficiently fleshed out.

We agree with the Reviewer that if connections existed among the different pathways, it would suggest a more intricate regulatory network controlling axon regeneration. We have attempted to

test this hypothesis but have not found compelling evidence yet that the *Irk1* and TNF signaling are directly linked. On the other hand, our existing and new data reveal a tight regulation of glia on axon regeneration, both positively and negatively, through the modulation of the expression and activation of the neuronal L-type calcium channels. The key point we hope to deliver is the balanced actions of three glia-neuron pathways to precisely control regeneration. The connection among the three pathways exists, at least, at the level of the L-type calcium channels that they modulate.

In addition, the link between TNF signaling and Ca- α 1D/Ca- β feels underdeveloped and would benefit from further characterization. The authors might consider to simplify the manuscript, by strengthening the relationship between Ca- α 1D/Ca- β and *Irk1*, while further developing the TNF part of the study in a separate manuscript.

We very much appreciate the suggestion. We have screened over 20 candidate genes, in order to delve deeper into the link between TNF signaling and Ca- α 1D/Ca- β , and found one additional component of the pathway – *Traf6* (Fig. 3). In our pursuit of the mechanistic links, we unexpectedly and excitingly found that the glia-neuron adenosine-AdoR-Ih axis acts as a negative regulator for axotomy-induced Ca²⁺ transients and axon regeneration. Therefore, we have refocused our story on this new route, which we propose may act antagonistically to the *Irk1* activity, in modulating the neuronal membrane properties essential for sustaining L-type calcium channel opening (Fig. 6 and 7). We believe that the concept of peripheral glia to inhibit and facilitate axon regeneration via three routes is significant and novel. We still think our strategy to include all three routes helps deliver this new concept and hope the Reviewer would find our approach acceptable.

In addition, the difference between C4da and C3da neurons is addressed in regards to their different response to Ca- α 1D and Ca- β expression and the generated Ca²⁺ spikes. It is unclear why there is a boost in regeneration for both cell types, but their spikes are very different (which is crucial as Ca²⁺ spikes are central to the message of the paper). Though the authors discuss this aspect, it should be further developed. This important part of the study is also dropped in favor of other lines of investigation. The difference in behavior of C4da and C3da neurons should be further investigated and explained.

This is an important point. Thank you for bringing this up. Regeneration cell type specificity is an emerging concept and has been seen in both flies and mammals (RGCs and DRGs). We agree our C4da and C3da neuron model is perfect for revealing the underlying mechanism, which has been rarely explored. We have now provided more data addressing this topic. Starting from the difference in the axotomy-induced Ca²⁺ transients, we first report the identification of the L-type calcium channels which mediate the Ca²⁺ transients and axon regeneration (Fig. 1). We then substantiated the expression analysis of the L-type calcium channels – with RNAScope and antibody staining, and show that the difference in the base levels as well as in the change of expression in response to injury (Fig. 2, Supplementary Fig. 2). We have thus identified the neuronal core machinery mediating regeneration cell type specificity. The more intriguing finding is that this core machinery is tightly modulated by glia via three routes. We further show that *wgn*, the receptor for the fly TNF *egr*, is differentially expressed between C4da and C3da neurons (Fig. 3l-n), and that loss-of-function and overexpression of *wgn* reduces C4da and increases C3da neuron axon regeneration, respectively (Fig. 3f-i). Moreover, we found that perturbing the adenosine-AdoR-Ih axis enhances axon regeneration in both C4da and C3da neurons (Fig. 6). We also postulate that it is possible that despite being wrapped by the same glial cell, the glial cytoplasm may selectively release TNF- α in the outpouchings surrounding C4da axons. Our work thus suggests the relationship between neurons and glia as a possible factor controlling subtype-specific regeneration and opens further avenues of exploration. This is discussed on P. 18.

Regarding Ca^{2+} spikes, we have revised the manuscript and decided to use Ca^{2+} transients to be more precise. Our data allow us to hypothesize that Ca^{2+} spikes per se may not be required to drive axon regeneration, but rather an elevated Ca^{2+} baseline with subthreshold transients may be sufficient to trigger downstream components that lead to regeneration, and that the calcium signal triggering regeneration may be malleable. This was similarly seen in *C. elegans*, where fluctuating Ca^{2+} transients, rather than spikes, promoted dendrite growth (Tao L, Coakley S, Shi R, Shen K. Dendrites use mechanosensitive channels to proofread ligand-mediated neurite extension during morphogenesis. *Dev Cell* 57, 1615-1629 e1613 (2022)). This is discussed on P. 18. The membrane properties of C4da neurons likely make them more likely to reach a threshold to produce Ca^{2+} spikes, and thus we still used spike quantification for C4da neurons which would reflect the underlying Ca^{2+} signals.

There are a few minor points, which should be addressed:

1. Throughout the manuscript there are inconsistencies regarding the order of appearance of the figure panels. For examples Figure 2h and 2i are mentioned in the text before figure 2c, g and j. A part of Figure 5a-c is discussed only at the end of the result sections, well after Figure 7 is introduced. The authors might want to rethink the figures, so that these inconsistencies are removed.

Thank you for the suggestion. We have reorganized the figures accordingly.

2. In Figure 6c there is no WT reference.

The WT trace is now added as a reference.

Reviewer #2:

In the manuscript "Coding axon regeneration by a glia-neuron ion channel module and TNF signaling" the authors follow up on previous work suggesting that neuronal calcium and glia control axon regeneration potential in *Drosophila* sensory neurons. In this manuscript they focus on a role for voltage-gated calcium channels (specifically Ca-a1D and Ca-beta) in neurons and an inward rectifying potassium channel in glia. They also show that TNF signaling between neurons and glia has some impact on axon regeneration. While there are some intriguing observations, the model has critical holes, and important controls are missing. It is therefore difficult to know whether the model they present in their last figure will hold up when these holes and controls are addressed.

Thank you for the critique. We have made our best efforts to address the gaps brought up below and hope the Reviewer would now find our conclusions more convincing.

The model they present in the abstract, title and final figure is that glial wrapping around the axon modulates Ca availability for entry through VGCCs through potassium buffering in glia. This model assumes that the VGCC is localized to the part of the cell that is wrapped by glia. They do not provide localization data for Ca-a1D, and the functional data from this study and other work suggests that it is abundant in dendrites, which are not wrapped by glia. In this work they show global calcium spikes throughout neurons, including in dendrites suggesting widespread distribution of Ca-a1D. A substantial dendritic population of Ca-a1D is supported by previous work showing Ca-a1D and Ca-beta function in dendrites after they have been disconnected from the cell body in pruning (1). The higher levels of Ca-a1D in neurons with large dendrites (Figure 1B) is also consistent with substantial presence in dendrites. Much of the Class IV dendrite arbor is hundreds of microns away from any glial wrap, as seen in Figure 3A, right column.

We thank the Reviewer for bringing up this point. We are very much aware of the dendrite pruning work from Emoto and colleagues. In order to address this question, we have performed the following experiments. We tested ablation of single or all dendrites of C4da neurons and found

that dendrite ablation does not result in Ca^{2+} transients. Only when axon is injured with or without dendrite ablation will there be obvious Ca^{2+} transients (Supplementary Fig. 1b-i). This highlights the requirement of the axon component. Using antibody staining, we found that both Ca- α 1D and Ca- β proteins are present at least in the C4da neuron soma (Supplementary Fig. 2). In particular, Ca- α 1D can be seen within the nerve bundles (Supplementary Fig. 2c). These results allow us to conclude that while Ca- α 1D is important for dendrite pruning, in the case of axotomy, the expression of Ca- α 1D and Ca- β in the soma and axon appears to be more relevant.

The manuscript also focuses extensively on the role of Ca spikes in axon regeneration, which they show are not important for improved regeneration in Class III neurons that express Ca- α 1D and Ca- β in Figure 2. It is therefore difficult to know whether focusing on spikes in the other figures is central to the biology in question.

This is a great point. Agreeing with the Reviewer, we have revised the manuscript and decided to use Ca^{2+} transients to be more precise. Our data allow us to hypothesize that Ca^{2+} spikes per se may not be required to drive axon regeneration, but rather an elevated Ca^{2+} baseline with subthreshold transients may be sufficient to trigger downstream components that lead to regeneration, and that the calcium signal triggering regeneration may be malleable. This was similarly seen in *C. elegans*, where fluctuating Ca^{2+} transients, rather than spikes, promoted dendrite growth (Tao L, Coakley S, Shi R, Shen K. Dendrites use mechanosensitive channels to proofread ligand-mediated neurite extension during morphogenesis. *Dev Cell* 57, 1615-1629 e1613 (2022)). This is discussed on P. 18. The membrane properties of C4da neurons likely make them more likely to reach a threshold to produce Ca^{2+} spikes, and thus we still used spike quantification for C4da neurons which would reflect the underlying Ca^{2+} signals. In C3da neurons, however, with overexpression of Ca- α 1D and Ca- β , the Ca^{2+} signal is unable to reach the threshold of producing spikes, but is sufficient to drive axon regeneration.

Critical missing controls

1. It is now standard and expected that antibody staining experiments include specificity controls. It seems appropriate to do the same for the RNAScope used extensively in multiple figures. The parallel controls to those expected for antibodies would be using the same RNA probes in knockdown cells or mutant animals. For example, for Figure 1 it would be good to knock down Ca- α 1D and Ca- β in neurons and show that the spots they quantitate go away. For figure 4, it is concluded that *lrrk1* spots are in glia- but the spots are on a bundle of axons and glia. A control of glial knockdown of *lrrk1* would strengthen this conclusion considerably.

We agree with the Reviewer that proper validation of the RNAScope result is critical and we have tried our best to address this concern. Here is our approach.

- RNAScope is a highly sensitive and specific *in situ* hybridization method. It utilizes a proprietary double Z probe design by Advanced Cell Diagnostics (ACD). It “employs a probe design strategy much akin to fluorescence resonance energy transfer (FRET), in which two independent probes (double Z probes) have to hybridize to the target sequence in tandem in order for signal amplification to occur. As it is highly unlikely that two independent probes will hybridize to a non-specific target right next to each other, this design concept ensures selective amplification of target-specific signals. For each target RNA species, ~20 double Z target probe pairs are designed to specifically hybridize to the target molecule, but not to non-targeted molecules.” From the literature that we surveyed, the conventional controls are a positive probe and a negative probe, which we have used in the manuscript. Here is just a short list of recent publications using the method for reference: Jiang Z et al., *Cell*. 2018 Oct 18;175(3):652-664.e12; Mi D et al., *Science*. 2018 Apr 6;360(6384):81-85; Szczot M et al., *Cell Rep*. 2017 Dec 5;21(10):2760-2771; Kim J et al., *Nat Neurosci*. 2016 Dec;19(12):1636-1646; Alexander GM et al., *Nat Commun*. 2016 Jan 25;7:10300).

- With that being said, we did test C4da neuron specific knockdown of Ca- α 1D and Ca- β , as suggested by the Reviewer. The results were not compelling though, as we did not see significant reduction of the signal, which likely is due to the limited knockdown efficiency of RNAi and the high sensitivity of the probes. Also need to point out that RNAi knockdown may have a more obvious effect on reducing protein level, rather than the level of mRNAs.
- We also considered the option of using mutants as suggested by the Reviewer. However, the mechanism of RNAScope to detect mRNAs makes it not feasible to use common mutants, as the mRNAs are still present. It will require a complete null allele, deleting the coding region or blocking transcription. In addition, given that hypomorphic mutants for Ca- α 1D, Ca- β or *Irk1* are already lethal at the embryonic or early larval stages, we would be not able to perform the experiments at the relevant larval stage. It would be possible to do Ca- α 1D, Ca- β by performing clonal analysis with MARCM, but no such alleles are available. In the case of *Irk1*, we are not yet aware of a robust way of performing MARCM in glial cells.
- Given all these considerations, we decided to validate our results using antibody staining as an alternative. We obtained a published polyclonal Ca- α 1D antibody, and generated our own polyclonal antibodies against Ca- α 1D, Ca- β or *Irk1*. Our results confirm that Ca- α 1D and Ca- β are present in C4da neurons, and *Irk1* is present in glial processes (Supplementary Fig. 2, Supplementary Fig. 5). We verified the specificity of these antibodies using RNAi knockdown. However, given that these are polyclonal antibodies and Ca- α 1D, Ca- β and *Irk1* are expressed in other surrounding tissues such as muscle cells, we have not been able to achieve subcellular resolution with the immunostaining.
- We hope that the Reviewer would appreciate our efforts attempting to address this issue and could find our updated results more acceptable.

2. In Figure 3 regeneration is compared with and without glial injury. Presumably cutting glia+axons involves considerably more tissue damage than cutting axons while leaving surrounding glia intact. The authors do not control for the added tissue damage in the glial injury condition so it is very difficult to interpret here (or in the paper they refer to) whether the key difference is the glial damage or amount of damage. In addition, the glial injury is not described in the Methods.

Thank you for the suggestion. We have performed extensive experiments to address this concern. We added a second method to specifically target the glial cell nucleus away from the axotomy site, to minimize tissue damage. We have also added a tissue injury control, to control the tissue damage associated with injuring axon+glia. The methods are now presented in Supplementary Fig. 3a. We found that injuring glial processes and glial nucleus injury produced similar results, by decreasing Ca²⁺ transients, while the tissue injury control is similar to the axotomy only condition (Fig. 3a-e, Supplementary Fig. 3b, c).

3. The effect of various knockdowns on cell shape before injury is not assayed, and so it is difficult to know whether the effect on regeneration is due to some large scale change in cell architecture (of neurons or glia) before injury, or whether it is a specific injury response defect. For example, in figure 3F the images for *Irk1* knockdown in glia look like the neurons have increased branching near the cell body. Is neuron shape altered at baseline in *Irk1* knockdown? Do glia still contact axons normally in *Irk1* knockdown? Similarly does *egr* or *wgn* knockdown affect neuronal or glia development or do they function specifically after injury as suggested?

Agreed. This is a critical control experiment. We have now performed morphology analyses of C4da neuron dendrite patterning and the glia wrap in glial knockdown of *egr*, *wgn*²² mutants and glial knockdown of *Irk1*. We found that none of the manipulations grossly alter the dendrite patterning or the wrapping area of C4d neurons (Supplementary Fig. 4b-e). The only difference we saw was a slight shift of the dendrite complexity (Supplementary Fig. 4f). We thus conclude

that *egr-wgn* and *Irkl* likely regulate axon regeneration independent of their modulation of neuron-glia morphology.

Additional points

It looks like there is data for two different types of class IV neurons in Figure 1F-I, although I could not find this specified anywhere. In the figure the blue bars are *ddaC*, but I am guessing that the data in 4I must be for a different class IV neuron as the regeneration index is quite different. It is not mentioned in the text why these two class IV neurons would have such different regeneration indices.

The difference of axon regeneration ability between *ddaC* and *v'ada* was reported in our original paper characterizing the axon regeneration capability of C4da neurons (Song et al., *Gene Dev* 2012). We have added the reference on P. 8. The regeneration ability of *ddaC* is a bit lower than *v'ada*, which may be due to their position and genetic program. We have also added the schematic diagram depicting the position of C4da and C3da neurons (Supplementary Fig. 1j-m).

An alternate interpretation for the membrane clumps in glia in S2: when the axon is cut, the glia that wrap them are also damaged- and these blobs are part of a repair pathway.

Thank you for the suggestion. We have added this alternative interpretation on P. 9.

In Figure 4 it looks like there is substantial damage to the glia in the images in A even though this is supposed to be an experiment where glia are not damaged. This raises a problem for quantitation of RNA in the wrapping glia. It looks like the area of the glial wrap is substantially reduced so the reduction in puncta shown in B is likely just due to reduction of glial area to examine.

This is a good point. We have changed our conclusion accordingly, as "The distribution of *Irkl* mRNA remained similar after axotomy, and enrichment was observed in glial processes surrounding or ahead of the axon tip (Fig. 5h, i, asterisk)." What we hope to deliver is that *Irkl* is present in the wrapping glia labeled by *nrv2-Gal4>CD4tdTomato*, with or without injury.

In figure 5D why is Rab5 used as a transcription reporter driven by *egr-Gal4*? Is it possible that endocytosis or something else affects Rab5 stability after injury?

We repeated the experiment using UAS-CD8GFP, which showed the same result (Supplementary Fig. 3d, e).

Figure 7 is somewhat confusing- the condition that rescues regeneration in Figure 5C (*egr3*, UAS-*Ca-a1D*) has comparable levels of *Ca1D* to the neighboring condition that does not regenerate (*wgn* RNAi). How does this fit with the model that *Ca-a1D* levels are a critical regulator of regeneration.

Our data show that both *egr³* mutants and C4da neuron specific knockdown of *wgn* reduce *Ca- α 1D* expression (Fig. 4f-h). Therefore, overexpression of *Ca- α 1D* in *egr³* mutants rescues the regeneration phenotype (Fig. 4i, j).

1. Kanamori T, Kanai MI, Dairyo Y, Yasunaga KI, Morikawa RK, Emoto K. Compartmentalized Calcium Transients Trigger Dendrite Pruning in Drosophila Sensory Neurons. *Science*. 2013. Epub 2013/06/01. doi: 10.1126/science.1234879. PubMed PMID: 23722427.

REVIEWER COMMENTS

Reviewer #1 (Remarks to the Author):

The manuscript submitted by Li and colleagues is a revised version from their first submission. The authors have managed to address my previous concerns. Importantly, the authors have reworked the flow of the manuscript so that the story feels less disjointed and added significant more molecular and mechanistic insights. In addition, the authors have modified some of their claims to better reflect the data and added needed controls.

I think this version of the manuscript is greatly improved and I believe it can be accepted for publication.

Reviewer #2 (Remarks to the Author):

Overall, the authors show that glial cells influence axon regeneration through controlling injury-induced calcium spiking. While the calcium spike and regeneration data support roles for voltage-gated calcium channels in neurons and neuron-glia signaling through *egr/wgn*, some of the other data is still low quality, poorly controlled and over-interpreted. They have enough interesting data without pushing some of the pieces of the story that are not so well supported. It would be helpful to finish filling in some of the holes (for example does *Caalpha1D* alone have an impact on Ca spiking) and maybe taking out some of the more peripheral pieces that do not add much to the story and could well be wrong (like ATP and direction of growth data in Figure 7a-d).

Figure 1. They show nicely that Ca transients occur 24h after axon injury in class IV neurons and control for this well. They also show that *Caalpha1D* is required for initiation of regeneration in these cells and correlate this with an effect on spiking.

Data in Figure 2 is still not very convincing. In the last round of reviews, it was suggested that they do some additional controls and while they tried quite valiantly to do them, the new data actually further undermines confidence in what is shown.

They say that *Caalpha1D* is higher in CIV than CIII neurons, but the beta subunit is the same in both cells. As all channels found in vivo are thought to contain alpha and beta subunit, it is unclear that these expression differences they note are meaningful. Their RNAscope data is further cast into doubt by the fact that the control experiment they were asked to do failed. In the wake of the reproducibility crisis, it is now standard to show that antibody signals are reduced or eliminated in RNAi or mutant samples. In their reviewer response they said that the RNAscope signal did not change in cells in which calcium channel subunit mRNAs were targeted with RNAi. This is very worrisome as these same RNAi hairpins give strong phenotypes in Figure 1. They say this is perhaps because RNAi reduces protein more than RNA, which does not make any sense as the hairpins target mRNA but protein must turn over for levels to be reduced so should be delayed compared to mRNA reduction. Their RNAscope data also does not agree with a reporter strategy based on a Gal4 insertion in *Cbeta* (Fig S2g and h). From the RNAscope data they conclude that the alpha and beta calcium channel subunit mRNAs are reduced after axon injury, and the beta is very strongly reduced in CIII neurons. The *Cbeta* reporter in Fig S2 goes up in CIV neurons after injury, and the condition they say changes the most in their RNAscope data (CIII after injury) is left out of this experiment for some reason. They further try to validate the RNAscope data using antibody staining Fig S2. Again, this would be great if the data looked good. However, the hazy signal in the images they show is not consistent with the antibody recognizing a membrane protein. They say that they see signal in the soma, dendrites and axons, but that is not evident from the images shown. They do not cross-validate their antibody staining and later overexpression experiments by showing that more Cav channels are present when they overexpress them.

Overall, their data in Figure 2 and S2 is not convincing, and the different ways to look at mRNA yielded conflicting results (RNAscope vs transcriptional reporter).

Their conclusion that the ratio of the alpha and beta subunit changes are important for channel properties is also problematic. They seem to imply that at baseline the alpha subunit may function without beta (line 150-156). The study they reference to support this is based on overexpression in *Xenopus* oocytes. However, the alpha subunit requires beta for trafficking to the plasma membrane, and alpha and beta are always isolated together from either muscle or neurons and form a tight complex, and most investigators believe that in vivo alpha always functions with beta. This background also makes one skeptical about the effects they see from overexpression of either subunit alone (Figure 2e). It would be helpful to have a control where they overexpress a different membrane protein to see whether boosting the secretory pathway in Class III neurons puts them into a growth state that allows more regeneration as it is difficult to believe that expressing alpha alone (where they see more of an effect than beta) would result in more functional channel at the plasma membrane as beta is required for trafficking. Alternately if they could use their antibody to show expression of alpha alone leads to more surface channel that would help. The data on whether they see additional Ca when both subunits are overexpressed is also not very convincing, and they do not show that alpha alone (or beta) which have an effect on regeneration also have an effect on Ca.

In summary, key additional experiments to shore up the overexpression piece in CIII include: a control to show that overexpression of a transmembrane protein (independent of it being a Ca channel) does not affect regeneration, and analysis of Ca in CIII neurons expressing alpha (and ideally beta) subunits alone, as the authors argue that these can also improve regeneration. The alpha subunit alone is particularly important as this is used again in Figure 4.

Figure 3: The additional tissue damage controls have now been performed, which is very helpful. The functional data on knockdown of *egr* and *wgn* is nice.

In 3J, the authors use *egr-Gal4* as a reporter of *egr* expression after injury. This is a great idea, but it is unclear why they chose *UAS-Rab5-GFP* as the thing that is driven by the *Gal4*. *Rab5* may well be regulated by injury post-translationally. Its trafficking could also be altered leading to increase in the area they quantitate rather than an increase in expression. It would be much better to use a GFP or RFP targeted to the nucleus (like they do with *Redstinger* in S2). The use of *Rab5* makes this experiment extremely difficult to interpret.

Figure 4: the connection between Ca and *egr/wgn* is very nicely made in 4a and b.

4f-h- from reading the legend it seems that this RNAscope experiment is from injured neurons. Is there also a baseline change in uninjured? Their section heading makes it sound like the effect is specific to injury, but without the uninjured data this isn't clear.

4i- as mentioned above, most of the data on Ca in Fig 2 is with both alpha and beta subunit expression, but here only alpha is used. It would be good to characterize the effect of alpha alone on Ca as that is the condition used here.

Figure 5 and 6- the data in these figures on effects of additional channels on Ca spiking and regeneration seem overall clear and broadly support the model that glia regulate Ca spikes in CIV and regeneration. However, I have to admit that the addition of so many new players was kind of confusing! And some of the individual pieces are difficult to make sense of: for example how does *Ih* RNAi increase regeneration, but reduce number of spiking neurons (while increasing spiking in the subset that do spike).

Figure 7. In a-c a reporter for ATP is used in glia. Is this normalized to another soluble fluorescent protein expressed in glia to take into account cell volume in the different regions? Moreover, this is intracellular ATP, so is there any reason to believe that this would translate into differential secretion of ATP in different regions, as opposed to say, just marking where glial metabolism is more active (or without the normalization to something else, that we are just looking at areas with more glial cytoplasm). E just seems like a repeat of what was shown in figure 6.

Minor points:

Figure S1- they say that they develop a new method of Calcium imaging, but do not describe what is different about what they do compared to other studies. It would be helpful if they could be more specific about what they changed and why.

Typo in y axis of graph in 7e

REVIEWER COMMENTS

Reviewer #1 (Remarks to the Author):

The manuscript submitted by Li and colleagues is a revised version from their first submission. The authors have managed to address my previous concerns. Importantly, the authors have reworked the flow of the manuscript so that the story feel less disjointed and added significant more molecular and mechanistic insights. In addition, the authors have modified some of their claims to better reflect the data and added needed controls. I think this version of the manuscript is greatly improved and I believe it can be accepted for publication.

We appreciate the Reviewer's support of our work!

Reviewer #2 (Remarks to the Author):

Overall, the authors show that glial cells influence axon regeneration through controlling injury-induced calcium spiking. While the calcium spike and regeneration data support roles for voltage-gated calcium channels in neurons and neuron-glia signaling through *egr/wgn*, some of the other data is still low quality, poorly controlled and over-interpreted. They have enough interesting data without pushing some of the pieces of the story that are not so well supported. It would be helpful to finish filling in some of the holes (for example does *Caalpha1D* alone have an impact on Ca spiking) and maybe taking out some of the more peripheral pieces that do not add much to the story and could well be wrong (like ATP and direction of growth data in Figure 7a-d).

We appreciate the Reviewer's suggestions to further solidify some of our hypotheses. In following their requests, our work now includes much more extensive controls to best support our conclusions.

Figure 1. They show nicely that Ca transients occur 24h after axon injury in class IV neurons and control for this well. They also show that *Caalpha1D* is required for initiation of regeneration in these cells and correlate this with an effect on spiking.

Thanks.

Data in Figure 2 is still not very convincing. In the last round of reviews, it was suggested that they do some additional controls and while they tried quite valiantly to do them, the new data actually further undermines confidence in what is shown.

They say that *Caalpha1D* is higher in CIV than CIII neurons, but the beta subunit is the same in both cells. As all channels found in vivo are thought to contain alpha and beta subunit, it is unclear that these expression differences they note are meaningful. Their RNAscope data is further cast into doubt by the fact that the control experiment they were asked to do failed. In the wake of the reproducibility crisis, it is now standard to show that antibody signals are reduced or eliminated in RNAi or mutant samples. In their reviewer response they said that the RNAscope signal did not change in cells in which calcium channel subunit mRNAs were targeted with RNAi. This is very worrisome as these same RNAi hairpins give strong phenotypes in Figure 1. They say this is perhaps because RNAi reduces protein more than RNA, which does not make any sense as the hairpins target mRNA but protein must turn over for levels to be reduced so should be delayed compared to mRNA reduction. Their RNAscope data also does not agree with a reporter strategy based on a Gal4 an insertion in *Cabeta* (Fig S2g and h). From the RNAscope data they conclude that the alpha and beta calcium channel subunit mRNAs are reduced after axon injury, and the beta is very strongly reduced in CIII neurons. The *Cabeta* reporter in Fig S2 goes up in CIV neurons after injury, and the condition they say changes the most in their RNAscope data (CIII after injury) is left out of this experiment for some reason. They further try to validate the RNAscope data using antibody staining Fig S2. Again, this would be great if the data looked good. However, the hazy signal in the images they show is not consistent with the antibody recognizing a membrane protein. They say that they see signal in the soma, dendrites and axons, but that is

not evident from the images shown. They do not cross-validate their antibody staining and later overexpression experiments by showing that more Cav channels are present when they overexpress them.

Overall, their data in Figure 2 and S2 is not convincing, and the different ways to look at mRNA yielded conflicting results (RNAscope vs transcriptional reporter).

We very much appreciate the Reviewer's thoughtful comments regarding the expression of the calcium channels. We have thus performed a complete new set of experiments detailed below, which we believe, fully address the concerns.

1. To fully address the Reviewer's concerns about RNAScope controls, in addition to the previous GFP and Bacterial RNA controls, we added positive and negative controls for *Ca- α 1D* and *Ca- β* . The positive control was performed via overexpression of the subunits in C3da neurons, showing greatly increased number of puncta in the soma (Supplementary Figure 2c, d).
2. For the negative control, we mentioned previously that C4da neuron RNAi knockdown did not yield a drastic reduction of the RNAScope signal. Thus, we combined the RNAis with the deficiency heterozygotes, in order to enhance the knockdown efficiency. With this new strategy, we found that knockdown for *Ca- α 1D* and *Ca- β* both lead to drastically decreased number of puncta in the soma (Supplementary Figure 2e-h).
3. For protein expression, we now include both positive and negative controls. The positive control was also performed using overexpression of individual channel subunits in C3da neurons. We found significantly increased immunostaining signal specifically in C3da neurons (Supplementary Figure 3g-j).
4. We included a better example of C4da neurons immunostained with the anti-*Ca- α 1D II* antibody, which clearly show the expression in the soma, axon and primary dendrites (Supplementary Figure 3k).
5. We agree with the Reviewer that the data generated with the Gal4 insertion in *Ca- β* is not compelling and have removed it. It is possible that the intron insertion of Gal4 in this case does not fully recapitulate the endogenous expression pattern of *Ca- β* .

These extensive controls further strengthened our conclusion.

Their conclusion that the ratio of the alpha and beta subunit changes are important for channel properties is also problematic. They seem to imply that at baseline the alpha subunit may function without beta (line 150-156). The study they reference to support this is based on overexpression in *Xenopus* oocytes. However, the alpha subunit requires beta for trafficking to the plasma membrane, and alpha and beta are always isolated together from either muscle or neurons and form a tight complex, and most investigators believe that in vivo alpha always functions with beta. This background also makes one skeptical about the effects they see from overexpression of either subunit alone (Figure 2e). It would be helpful to have a control where they overexpress a different membrane protein to see whether boosting the secretory pathway in Class III neurons puts them into a growth state that allows more regeneration as it is difficult to believe that expressing alpha alone (where they see more of an effect than beta) would result in more functional channel at the plasma membrane as beta is required for trafficking. Alternately if they could use their antibody to show expression of alpha alone leads to more surface channel that would help. The data on whether they see additional Ca when both subunits are overexpressed is also not very convincing, and they do not show that alpha alone (or beta) which have an effect on regeneration also have an effect on Ca.

In summary, key additional experiments to shore up the overexpression piece in CIII include: a control to show that overexpression of a transmembrane protein (independent of it being a Ca channel) does not affect regeneration, and analysis of Ca in CIII neurons expressing alpha (and

ideally beta) subunits alone, as the authors argue that these can also improve regeneration. The alpha subunit alone is particularly important as this is used again in Figure 4.

Thank you for the insightful comments. We have performed all the suggested experiments, which support our conclusions.

1. Regarding the concern of general overexpression of a transmembrane protein boosting axon regeneration, we have tested overexpression of the TrpA1 channel at the non-activating temperature and did not see enhanced axon regeneration (Supplementary Figure 2i, j). This indicates that the increased axon regeneration after overexpression of Ca- α 1D and Ca- β is specific.
2. Regarding the concern of surface expression, Ca- α 1D II antibody staining was done both with non-permeabilized (which better reflects surface expression) and permeabilized conditions. We show that Ca- α 1D overexpression alone in C3da neurons leads to more surface channel expression (Supplementary Figure 3g).
3. We show that overexpression of Ca- α 1D or Ca- β alone in C3da neurons indeed slightly increases the subthreshold transients (STT), but to a lesser extent than the co-overexpression (Supplementary Figure 2k-m).

Figure 3: The additional tissue damage controls have now been performed, which is very helpful. The functional data on knockdown of *egr* and *wgn* is nice. Thanks.

In 3J, the authors use *egr-Gal4* as a reporter of *egr* expression after injury. This is a great idea, but it is unclear why they chose UAS-Rab5-GFP as the thing that is driven by the Gal4, Rab5 may well be regulated by injury post-translationally. Its trafficking could also be altered leading to increase in the area they quantitate rather than an increase in expression. It would be much better to use a GFP or RFP targeted to the nucleus (like they do with Redstinger in S2). The use of Rab5 makes this experiment extremely difficult to interpret.

We agree with the Reviewer and have included both the UAS-Rab5-GFP and the general UAS-mCD8GFP reporters, which both reflected increased expression after injury (Supplementary Figure 4d, e).

Figure 4: the connection between Ca and *egr/wgn* is very nicely made in 4a and b. 4f-h- from reading the legend it seems that this RNAscope experiment is from injured neurons. Is there also a baseline change in uninjured? Their section heading makes it sound like the effect is specific to injury, but without the uninjured data this isn't clear.

We have included the data for the uninjured condition, which show that *egr*³ mutants do not affect the expression of Ca- α 1D or Ca- β , while C4da neuron *wgn* knockdown increases their expression (Supplementary Figure 4f, g). This confirms that the reduction of Ca- α 1D after LoF of *egr* or *wgn* is specific to the injury. Further study of the base line *wgn* expression change in the various conditions is warranted, but is beyond the scope of the current paper.

4i- as mentioned above, most of the data on Ca in Fig 2 is with both alpha and beta subunit expression, but here only alpha is used. It would be good to characterize the effect of alpha alone on Ca as that is the condition used here.

As mentioned above, we have performed Ca²⁺ imaging on C3da neurons overexpressing Ca- α 1D or Ca- β alone and found they indeed slightly increases the subthreshold transients (STT), but to a lesser extent than the co-overexpression (Supplementary Figure 2k-m).

Figure 5 and 6- the data in these figures on effects of additional channels on Ca spiking and regeneration seem overall clear and broadly support the model that glia regulate Ca spikes in CIV

and regeneration. However, I have to admit that the addition of so many new players was kind of confusing! And some of the individual pieces are difficult to make sense of: for example how does lh RNAi increase regeneration, but reduce number of spiking neurons (while increasing spiking in the subset that do spike).

We appreciate the Reviewer's overall support of these sections and also the concern. We believe that the addition of these data substantially strengthens the story, depicting a wholistic view of the channel network underlying glia-neuron interaction during axon regeneration. We hope our schematic diagram helps clarify the roles of the various players. Regarding the lh RNAi, notably the decrease in % of neurons showing spiking is not statistically significant. We have added a discussion on P. 15 "lh knockdown not increasing the percentage of spiking neurons suggests the channels play a role only in attenuating the Ca²⁺ spikes of neurons with sufficient Ca- α 1D and Ca- β levels and does not affect neurons incapable of spiking".

Figure 7. In a-c a reporter for ATP is used in glia. Is this normalized to another soluble fluorescent protein expressed in glia to take into account cell volume in the different regions? Moreover, this is intracellular ATP, so is there any reason to believe that this would translate into differential secretion of ATP in different regions, as opposed to say, just marking where glial metabolism is more active (or without the normalization to something else, that we are just looking at areas with more glial cytoplasm). E just seems like a repeat of what was shown in figure 6.

This is a great point! The data was normalized to the first region of interest in order to compare the trend between data sets. To address the concern about cell volume, we compared this to glial expression of mRFP, which showed no trend in either direction (Supplementary Figure 8a, b), suggesting that the gradient of the ATP sensor is not due to glial volume or location. We agree that an ATP sensor sensitive enough to measure extracellular ATP will help further clarify the secretion of ATP from glia, but it is beyond the scope of the current study. The data in Figure 7e is different from Figure 6 in that the first graph is a percentage of axons which regenerated in one direction or the other, while the second set quantifies the length growth in either direction and expresses it as a ratio, as reported previously (Wang et al., 2020). We have included the detailed quantification in the Methods.

Minor points:

Figure S1- they say that they develop a new method of Calcium imaging, but do not describe what is different about what they do compared to other studies. It would be helpful if they could be more specific about what they changed and why.

Thanks for the suggestion. The specific improvements were that it was both *in vivo* and unanesthetized. We have update this on P. 5 to better reflect those key points.

Typo in y axis of graph in 7e.

Fixed the typo.

REVIEWER COMMENTS

Reviewer #2 (Remarks to the Author):

The additional controls, particularly for RNAscope data, help strengthen the manuscript. However, there are still some issues with the text that are important to resolve.

In the introduction it is stated "there are no studies investigating the electrical effect of glia on regenerating axon," but there is a paper on glial cells acting upstream of Ca spikes in injured neurons and axon regeneration now:

<https://www.sciencedirect.com/science/article/pii/S153458072300103X?via=ihub>

so this statement needs to be removed.

A similar statement is made in the discussion: "To our knowledge, this is the first demonstration of glia controlling neural activity after injury to regulate regeneration."

The statement "However, in injured C4da neurons, Ca- α 1D was less severely reduced, and Ca- β appeared largely unaltered" does not seem to be well-supported by the data. I do not see a comparison between the reduction in Ca- α 1D in C3da and C4da neurons in figure 2a to show a difference in the amount of reduction between the two cell types.

As mentioned in the previous round of review the arguments around ratios between the alpha and beta subunits are very tenuous and do not make biological sense. If the ratio is important, shouldn't the injured C3 and C4 neurons have similar calcium responses as the ratios in both injured cell types is very similar based on the data in Figure 2C? Moreover, in new data in Figure S2m Ca-beta is shown to have a similar effect to alpha, so if the argument is that you need high alpha relative to beta then this does not make sense. Alternately if the argument is that beta needs to be higher (based on lack of reduction in beta in C4) then the alpha effect does not make sense. Cell-surface channels consist of a single alpha and single beta subunit, and unincorporated subunits should not exit the ER. How do the authors imagine different ratios influencing function? This is not addressed in the Results and the issue of ratios is largely dropped in the discussion.

It is very unclear how to fit any reduction of VGCC RNA into a story about differing channel levels between cell types influence the response to injury (if Ca spikes promote regeneration, why is the RNA downregulated?), and the wording and presentation of the data in the figure seems to push a confusing and weak point. It seems much more important to point out that before and after injury alpha subunit RNA level is higher in C4 than C3. Though of course, really what is relevant is surface protein. Protein data is in S3 and the staining is not super convincing. For example, in the injury image in S3A the alpha subunit staining is lower in every cell in the image compared to uninjured so it is unclear that the reduction shown in the graph is at all specific to the injured neuron. Moreover, the absolute levels compared between C3 and C4 indicate a quite subtle difference in expression (C3 looks like it has between 85 and 90% of C4, and C4 is a much larger cell so presumably more channel would be needed to have the same effect).

The additional controls in the section on egr and wgn help strengthen this section.

It might be helpful to put the Irk data in context of the paper noted above as it shows that glia also have calcium spikes.

The data in Figure 6A and B shows that AdoR reduction increases regeneration. This is in direct conflict with data in the paper noted above that show AdoR expression is required for regeneration in C4da neurons.

In the previous response the authors mention that they use a plasma membrane marker as a volume control for ATP measurements. A surface marker does not report on volume so I am confused by this statement.

The discussion is quite speculative, but misses essential points like how their data showing AdoR inhibits regeneration can be reconciled with the recently published data in Dev Cell showing that AdoR promotes regeneration in the same cell type.

The issues surrounding VGCC levels after injury are also made more confusing in the discussion. For example this statement "The question of why WT C3da neurons do not modulate their Ca- β /Ca- α 1D levels after injury is not fully answered" does not fit with the data in Figure 2 showing that C3da neurons modulate Cabeta and alpha MORE than c4da (Figure 2b) after injury. The discussion should be rewritten to more accurately reflect what is shown in the Results and also take into account the recent Dev Cell paper.

Reviewer #2 (Remarks to the Author):

The additional controls, particularly for RNAscope data, help strengthen the manuscript.
Thank you.

However, there are still some issues with the text that are important to resolve.

In the introduction it is stated “there are no studies investigating the electrical effect of glia on regenerating axon,” but there is a paper on glial cells acting upstream of Ca spikes in injured neurons and axon regeneration now:

<https://www.sciencedirect.com/science/article/pii/S153458072300103X?via=ihub>

so this statement needs to be removed.

Thank you for pointing this out. We have changed it into “While many biochemical signaling mechanisms between neurons and glia have been identified, how the electrical effect of glia influences regenerating axons is less well studied.”

A similar statement is made in the discussion: “To our knowledge, this is the first demonstration of glia controlling neural activity after injury to regulate regeneration.”

We have removed it from the Discussion.

The statement “However, in injured C4da neurons, Ca- α 1D was less severely reduced, and Ca- β appeared largely unaltered” does not seem to be well-supported by the data. I do not see a comparison between the reduction in Ca- α 1D in C3da and C4da neurons in figure 2a to show a difference in the amount of reduction between the two cell types.

Sorry about the confusion. We have changed it into “In injured C4da neurons, Ca- α 1D was also reduced but significant amount remained, whereas Ca- β appeared largely unaltered”.

As mentioned in the previous round of review the arguments around ratios between the alpha and beta subunits are very tenuous and do not make biological sense. If the ratio is important, shouldn't the injured C3 and C4 neurons have similar calcium responses as the ratios in both injured cell types is very similar based on the data in Figure 2C? Moreover, in new data in Figure S2m Ca-beta is shown to have a similar effect to alpha, so if the argument is that you need high alpha relative to beta then this does not make sense. Alternately if the argument is that beta needs to be higher (based on lack of reduction in beta in C4) then the alpha effect does not make sense. Cell-surface channels consist of a single alpha and single beta subunit, and unincorporated subunits should not exit the ER. How do the authors imagine different ratios influencing function? This is not addressed in the Results and is the issue of ratios is largely dropped in the discussion.

We appreciate this comment. We have now extensively discussed our ratio hypothesis in the Discussion (P. 18) and added a schematic diagram (Supplementary Fig. 3l, m) based on our interpretation of the data. Specifically, here is our rationale:

1. Our data suggest that neurons also need to express a minimum level of both Ca- α 1D and Ca- β to regenerate axons. Thus, both the expression level of Ca- α 1D and Ca- β , and the Ca- β /Ca- α 1D ratio are critical for the axotomy-induced Ca²⁺ transients and axon regeneration. In injured C3da neurons, the expression level of Ca- α 1D and Ca- β is much lower than that of C4da neurons, which is the reason why they fail to show Ca²⁺ transients and axon regeneration, even with a proper ratio. This is further supported by the reduced Ca²⁺ transients and axon regeneration of C4da neurons when the egr-wgn axis is perturbed, which significantly reduces Ca- α 1D expression after injury.
2. In Supplementary Fig. 2k-m, we show overexpression of Ca- β or Ca- α 1D increases STT in C3da neurons to 35% or 50%, respectively, compared to 60% resulting from double

overexpression. This is in full agreement with the axon regeneration data. Our interpretation is that, again, you need sufficient levels of *Ca-α1D* plus an optimal ratio to achieve most significant Ca^{2+} transients and regeneration. We depicted the various scenarios in C4da and C3da neurons, based on our interpretation, in Supplementary Fig. 3l, m, and below. In particular, in C3da neurons, overexpression of *Ca-β* or *Ca-α1D* will both increase the number of channels of an optimal configuration (optimal ratio or activation states, see below), compared to WT. *Ca-α1D* overexpression is slightly better than *Ca-β* overexpression, as there may be more *Ca-α1D* even without the optimal configuration. They may still produce Ca^{2+} to some extent, despite sub-optimally. After all, we propose that *Ca-β* helps to de-inactivate *Ca-α1D*, to allow sustainable Ca^{2+} transients.

3. Thanks for bringing up the “a single alpha and single beta subunit” concept. We totally agree that this is a well-recognized concept – interaction of *Ca-α1D* and *Ca-β* as being important for ER trafficking to the membrane. We thus have discussed this from two perspectives – stoichiometry (a-c) and activation states (d).
 - a. There are examples in the literature showing that pore-forming subunits are capable of being targeted to the plasma membrane in the absence of its beta subunit, under certain circumstances. Proteasome inhibition allowed for Cav1.2’s targeting to the plasma membrane in the absence of beta (Altier et al., 2010). It was also shown that calcium alpha-beta interaction is reversible and that the beta subunits function as regulatory proteins rather than stoichiometric subunits (Hidalgo et al., 2006).

Altier, C., Garcia-Caballero, A., Simms, B. et al. The Cavβ subunit prevents RFP2-mediated ubiquitination and proteasomal degradation of L-type channels. *Nat Neurosci* 14, 173–180 (2011).

Hidalgo P, Gonzalez-Gutierrez G, Garcia-Olivares J, Neely A. The alpha1-beta-subunit interaction that modulates calcium channel activity is reversible and requires a competent alpha-interaction domain. *J Biol Chem*. 2006 Aug 25;281(34):24104-10.
 - b. Beta subunits are also capable of leaving the ER in the absence of a pore-forming subunit and the beta subunits ER exiting motif was previously identified (Fang and Colecraft, 2011).

Fang K, Colecraft HM. Mechanism of auxiliary β -subunit-mediated membrane targeting of L-type (Ca(V)1.2) channels. *J Physiol.* 2011 Sep 15;589(Pt 18):4437-55.

- c. These findings were reviewed by Simms and Zamponi. It is possible that other changes in the regenerative context allow such trafficking to be more common. This is an area that is worth further investigations.

Simms, B.A., Zamponi, G.W. Trafficking and stability of voltage-gated calcium channels. *Cell. Mol. Life Sci.* 69, 843–856 (2012).

- d. Regarding our ratio hypothesis: we expect different ratios to primarily impact ionic currents of Ca- α 1D downstream of the ER, similarly postulated by Neely and colleagues, who hypothesized a two-state model in which a second beta subunit binds after ER release and is able to modulate channel activity. It is also suggested that the essential Cav β modulatory properties are independent of the AID (α 1-interacting domain in the I-II intracellular linker), as has been proposed as the primary interaction site in α 1 subunits (Maltez et al., 2005). Therefore, alternatively, it is plausible that the effect we saw during regeneration is contributed by Ca- β binding to Ca- α 1D to reach the second activation state.

Maltez, J., Nunziato, D., Kim, J. et al. Essential Cav β modulatory properties are AID-independent. *Nat Struct Mol Biol* 12, 372–377 (2005).

Neely A, Garcia-Olivares J, Voswinkel S, Horstkott H, Hidalgo P. Folding of active calcium channel beta(1b) -subunit by size-exclusion chromatography and its role on channel function. *J Biol Chem.* 2004 May 21;279(21):21689-94.

We have also added a few caveats in the Discussion, thanks to the Reviewer's feedbacks.

1. A similar ratio of mRNA does not necessitate an equivalent amount of surface protein. We caution that more in-depth analyses in multiple systems are warranted to further substantiate the ratio hypothesis.
2. We state that we propose the ratio hypothesis based on our interpretation of the data, but we do not preclude other alternative possibilities.

It is very unclear how to fit any reduction of VGCC RNA into a story about differing channel levels between cell types influence the response to injury (if Ca spikes promote regeneration, why is the RNA downregulated?), and the wording and presentation of the data in the figure seems to push a confusing and weak point. It seems much more important to point out that before and after injury alpha subunit RNA level is higher in C4 than C3. Though of course, really what is relevant is surface protein. Protein data is in S3 and the staining is not super convincing. For example, in the injury image in S3A the alpha subunit staining is lower in every cell in the image compared to uninjured so it is unclear that the reduction shown in the graph is at all specific to the injured neuron. Moreover, the absolute levels compared between C3 and C4 indicate a quite subtle difference in expression (C3 looks like it has between 85 and 90% of C4, and C4 is a much larger cell so presumably more channel would be needed to have the same effect).

Thanks for this stimulating question – why the downregulation of *Ca- α 1D* RNAs. More intriguingly, why the differential change of *Ca- α 1D* and *Ca- β* RNAs in different types of neurons. After injury, downregulation of a number of important factors occurs, and similarly, upregulation of anti-regenerative factors occurs. The specific reason for differences in RNA regulation would be an important question for follow-up studies. With our response above, we hope we have now

made it clearer that it is both the absolute levels of the channel subunits and their ratio that are critical for the regenerative response.

We explicitly state that our ratio hypothesis is largely based on the RNA level studies. We also agree that more precise metrics of surface protein levels will help better understand the mechanics between channel proteins during regeneration, but would be beyond the scope of the current study. We hope to convey that our RNA and protein data corroborate one another, demonstrating differential regulation of subunits between C3da and C4da after injury, resulting in different axon regeneration.

In Supplementary Fig. 3A, we believe that the reduction of Ca- α 1D staining in neighboring neurons is because their axons were also injured.

We agree that the difference in Ca- α 1D immunostaining between C4da and C3da neurons were not as striking, but indeed statistically significant. We believe this is likely contributed by the limited sensitivity of the antibody and the wide-expression of the protein. We think this is at least consistent with our RNAScope data, which is more specific and sensitive. As we have postulated, it is the absolute level and the ratio that is important.

We added the following sentence to help clarify. "As noted, Ca- α 1D OE in C4da neurons impaired regeneration, further suggesting that it is not only the quantity, but also the ratio that is needed for optimized function, a concept which has been explored in many biological contexts – molecular titration (Babu et al., 2011).

Babu MM, van der Lee R, de Groot NS, Gsponer J. Intrinsically disordered proteins: regulation and disease. *Curr Opin Struct Biol.* 2011 Jun;21(3):432-40.

The additional controls in the section on *egr* and *wgn* help strengthen this section.

Thanks!

It might be helpful to put the *Irk* data in context of the paper noted above as it shows that glia also have calcium spikes.

Thank you for the suggestion. At this point, we are not fully convinced by the data from the other paper, see our response below.

The data in Figure 6A and B shows that AdoR reduction increases regeneration. This is in direct conflict with data in the paper noted above that show AdoR expression is required for regeneration in C4da neurons.

We noticed that the other paper, as referred by the Reviewer, claimed that adenosine signaling promotes axon regeneration, which is opposite to what we have observed. 1. They showed that axon regeneration is reduced in an adenosine receptor (AdoR) loss-of-function (LoF) mutant (*their fig. 2l*). However, they only examined one mutant allele, and more importantly the regeneration index from their Control (~0.75) is much elevated than the Control in any other figures (~0.5), while the AdoR^{-/-} value is in fact ~0.5. In our case, we have examined two AdoR LoF mutant alleles and two independent AdoR neuronal RNAs, none of which show any sign of regeneration reduction. On the contrary, we saw over 50% increase of regeneration index throughout (Fig. 6). We have also demonstrated the genetic interaction between AdoR and *Ih*, and LoF of *Ih* also promotes regeneration (Fig. 6). 2. They reported that overexpression of AdoR leads to enhancement of axon regeneration (*their fig. 2k*), which we also saw. But we believe, based on the more convincing LoF data, the "gain-of-function (GoF)" experiment is not as informative. We speculate that the dose of AdoR and thus adenosine signaling is critical. AdoR overexpression may disrupt its normal localization or function, thus leading to a LoF phenotype. They also looked at retinal ganglion cell (RGC) axon regeneration after AdoR overexpression and found an enhancement (*their fig. 7*). This may be potentially confounded by the overexpression. In contrast, our data show that adenosine treatment *in vitro* inhibits

mammalian axon regeneration (Fig. 7), consistent with our fly data. 3. They looked at AdoR expression in fly sensory neurons with the AdoR-Gal4 reporter and claimed that it is only expressed in the regenerative C4da neurons, but not the non-regenerative C3da neurons (*their fig. 2G*). However, using the same reagent, we also saw expression in C3da neurons albeit weaker. Moreover, with antibody staining, the protein level of AdoR is comparable between C4da and C3da neurons (Fig. 6). The strength in our study is the consistency seen across multiple alleles (mutants and knockdowns), and across species. Therefore, we have cited the other paper in our Discussion, but kept it open for future studies to address the discrepancy (P. 17).

In the previous response the authors mention that they use a plasma membrane marker as a volume control for ATP measurements. A surface marker does not report on volume so I am confused by this statement.

Sorry about the confusion. We used the mRFP mainly as a control to show the specificity of iATP. While it is a membrane marker, it can also mark the cytoplasmic compartment. From the mRFP signal, we aim to show that there is no gradient either on the membrane or within the cytoplasm. This is further substantiated by the *egr-Gal4>Rab5-GFP* reporter, which is not a plasma membrane marker. We have updated the text to clarify.

The discussion is quite speculative, but misses essential points like how their data showing AdoR inhibits regeneration can be reconciled with the recently published data in Dev Cell showing that AdoR promotes regeneration in the same cell type.

We have now substantiated the discussions, incorporating the helpful suggestions from the Reviewer. As mentioned above, we have cited the other paper in our Discussion, but kept it open for future studies to address the discrepancy.

The issues surrounding VGCC levels after injury are also made more confusing in the discussion. For example this statement “The question of why WT C3da neurons do not modulate their Ca-β/Ca-α1D levels after injury is not fully answered” does not fit with the data in Figure 2 showing that C3da neurons modulate Cabeta and alpha MORE than c4da (Figure 2b) after injury. The discussion should be rewritten to more accurately reflect what is shown in the Results and also take into account the recent Dev Cell paper.

Sorry about the confusion. By “modulate” we meant modulate in a manner that promotes regeneration. We have reworded this to be clearer, “The question of why WT C3da neurons do not change their Ca-β/Ca-α1D levels to promote regeneration after injury is not fully answered.” As responded above, we have substantially updated the Discussion, cited the other paper, but kept it open for future studies to address the discrepancy.

REVIEWERS' COMMENTS

Reviewer #2 (Remarks to the Author):

I thank the authors for addressing my concerns. The context and thought process is now much clearer. In particular the additions to the discussion help make sense of the data and how the authors think about it. I think it also helps to have addressed the other paper on AdoR directly in the discussion.